# A zebrafish model of chronic heart failure caused by protein aggregation in heart valves

Yitong Li[1], Shingo Maegawa[2], Ryo Kimura [1,3,4,5 ✉], Shiho R. Suzuki [3], Taiki Nishimura[1] & Masatoshi Hagiwara [1]

We previously developed a unique zebrafish, *tomato*, expressing the red fluorescent protein DsRed under the control of the *tph2* promoter. Unexpectedly, the *tomato* fish showed cardiac dilation and symptoms resembling those of chronic heart failure. Therefore, we investigate the pathogenesis and causes of these cardiac abnormalities. Crossbreeding suggests a link between DsRed expression and heart enlargement. Histological analysis confirms atrial dilation and thickening of the atrioventricular valve. RNA sequencing and immunohistological imaging reveal that valve thickness is due to the upregulated inflammation, cell proliferation, and epithelial-mesenchymal transition. Importantly, immunohistological staining elucidates DsRed accumulation in the atrioventricular valve, and removal of DsRed via Cre-mRNA injection rescues these heart defects. This study establishes a potential model of heart failure caused by protein aggregation in the cardiac valve. These findings highlight the potential of this transgenic zebrafish for investigating valvular heart diseases and chronic heart failure and developing new therapies.

Chronic heart failure (CHF) is a terminal condition resulting from various heart diseases, including cardiac structural abnormalities or insufficient myocardial function, which eventually lead to systemic inadequate blood supply[1]. Among these underlying causes, valvular cardiac disease is a significant contributor, primarily by causing progressive volume and pressure overload that compromises cardiac output performance[2–4]. This maladaptive process is characterized by a protracted and progressive trajectory, often taking years to transition from the compensated cardiac function to overt heart failure[5]. Despite significant clinical relevance, the molecular and structural changes driving valvular dysfunction-induced heart failure remain poorly understood[5,6]. Establishing appropriate animal models that mimic valve-induced cardiac pathology is essential for elucidating the mechanisms of CHF progression and for evaluating potential therapeutic interventions.

Recent significant breakthroughs have been made in various aspects of heart research using the zebrafish model, including heart development, heart regeneration, and the pathogenesis of heart diseases[7–9]. The advantages of the transparent bodies and rapid cardiac development capacity of zebrafish larvae have allowed the development of heart failure models at the larval stage through gene editing or drug induction[10,11]. Although useful,

these models are limited as they validate only one component of the pathophysiological mechanisms of heart failure, such as cardiac hypertrophy, dysfunction, or remodeling. More importantly, these models are not progressive. In particular, models that induce heart failure at 3 days postfertilization (dpf) fail to reproduce the long-term progression of this disease[12,13]. Therefore, a new heart failure model that can more fully represent progressive cardiopathological changes is urgently needed.

Herein, we introduce a progressive transgenic zebrafish model that expresses the red fluorescent protein DsRed in the cardiac valve, which leads to multiple cardiac phenotypes. Heart enlargement was observed during the larval stage (30 dpf) and became more pronounced with age. Cardiovascular phenotypic diagnosis revealed that this zebrafish model was associated with symptoms consistent with heart failure. Histological examination demonstrated atrial dilation resulting from atrioventricular (AV) valve thickening, which restricted valve motion and caused blood regurgitation. Further investigations into the pathogenesis of AV valve thickness showed that it resulted from DsRed accumulation and subsequent upregulation of epithelial-to-mesenchymal transition (EMT) and cell proliferation on the AV valve. These findings suggest that cardiac valve remodeling can be attributed to the cytotoxic effects of DsRed aggregates, which progressively

[1]Department of Drug Discovery Medicine, Graduate School of Medicine, Kyoto University, Kyoto, Japan. [2]Department of Intelligence Science and Technology, Graduate School of Informatics, Kyoto University, Kyoto, Japan. [3]Department of Child Development, United Graduate School of Child Development, The University of Osaka, Osaka, Japan. [4]Department of Psychiatry, Graduate School of Medicine, The University of Osaka, Osaka, Japan. [5]Bioinformatics Center, Research Institute for Microbial Diseases, The University of Osaka, Osaka, Japan. ✉ e-mail: kimura.ryo@ugscd.osaka-u.ac.jp

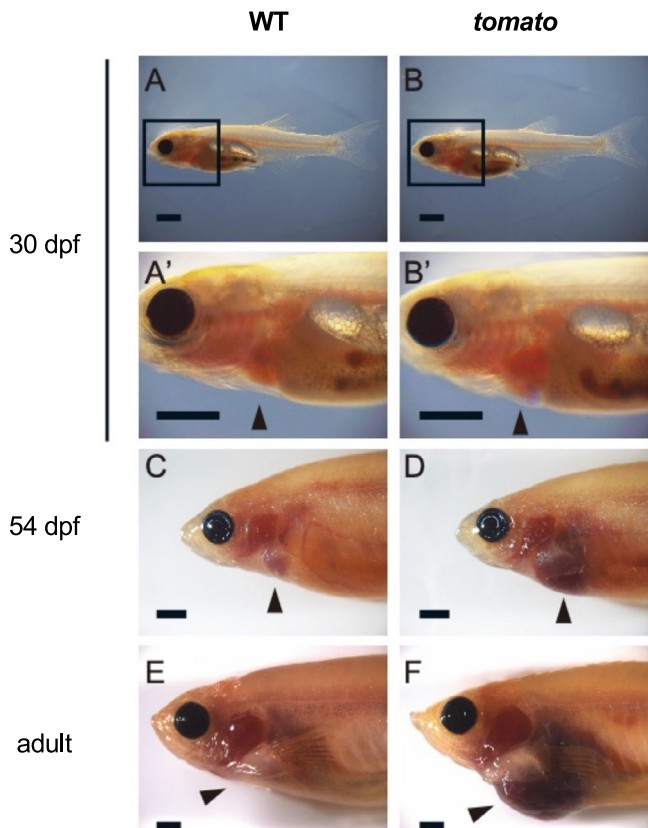

WT *tomato*

30 dpf

54 dpf

adult

**Fig. 1 | *Tomato* zebrafish show heart enlargement with growth. A through F,** Bright-field images of both wild-type (WT) and *tomato* zebrafish at 30 days post-fertilization (dpf) (**A, A', B, B'**), 54 dpf (**C, D**), and adulthood (**E, F**). **A'** and **B'**, Enlarged views of **A, B**. Scale bar, 1 mm. Black arrows: heart position.

lead to heart failure. The results of our study provide new insights into valvular disease-induced heart failure and offer new ideas for future research on valvular function and its potential progression to heart failure.

## Results

### Generation and identification of the *tomato* transgenic line

We have identified an upstream region of tryptophan hydroxylase 2 (*tph2*) ~5 kbp in length as a specific promoter/enhancer for serotonergic neurons (Supplementary File 1). An artificial construct (Supplementary Fig. 1A) was engineered for optogenetic manipulation of serotonergic neurons. The DsRed gene in this transgene served as a marker to confirm successful insertion. As expected, transgenic fish carrying this artificial gene exhibited DsRed expression in the dorsal raphe nucleus, in a pattern consistent with *tph2* expression detected by whole-mount in situ hybridization (ISH) (Supplementary Fig. 1B). The transgenic fish line was termed *tomato*. No distinct phenotypes were observed in *tomato* during the embryonic phase (approximately 72 h post-fertilization [hpf]). However, as *tomato* fish aged, they unexpectedly developed expanded hearts (Fig. 1). In a breeding experiment, *tomato* fish were crossed with wild-type (WT) fish, resulting in offspring with a 1:1 phenotypic ratio, in accordance with Mendelian inheritance. Moreover, all *tomato* fish exhibited heart enlargement, while none of the WT fish showed this phenotype, confirming that the transgenic insertion is tightly associated with the expanded heart phenotype (Supplementary Fig. 1C).

### Survival rate and physiological characteristics of *tomato* zebrafish

To investigate the developmental progression of heart formation in *tomato* zebrafish, we first examined their cardiac phenotypes. During zebrafish husbandry, a higher frequency of death was observed in the *tomato* fish.

Nearly 95.5% of WT fish survived at 2 months post-fertilization (mpf), whereas only 73.3% of *tomato* fish survived (Fig. 2A), indicating a significantly higher mortality rate in *tomato* fish. Combined with the observation of a noticeable enlargement in cardiac morphology in *tomato* fish, we suspected that such heart remodeling probably progressed to decompensation and developed into CHF in adulthood. To verify this idea, we evaluated several cardiac function parameters, particularly for symptoms of heart failure. Among these, heart rate serves as an indicator of the overall cardiac function. We analyzed heart rate changes at different developmental stages and found no significant difference in heart rate at the larval period (3 dpf) and juvenile stage (2 mpf) (Supplementary Fig. 2A, B). Although the result indicated a significant decrease in adult (3 mpf) *tomato* fish, averaging 91 beats per minute, compared with 104 beats per minute in WT fish (Fig. 2B). This was consistent with previous findings indicating lower heart rate in heart failure models of zebrafish[14–16].

Natriuretic peptide B (NPPB), also known as BNP, commonly serves as a sensitive diagnostic biomarker of CHF that responds to pressure and volume overload[17]. Through quantitative PCR (qPCR), the expression of *nppb* in adult *tomato* fish was significantly increased, being 2.15 times higher than that of WT (Fig. 2C).

Echocardiography was performed to detect heart function and blood flow in the *tomato* fish (Fig. 2D). Unlike in healthy humans, the WT zebrafish atrioventricular pulse-wave Doppler signal is composed of a low E peak and a dominant A peak (Fig. 2E). This result was consistent with those previously reported[18]. A downstream regurgitation wave was observed in *tomato* zebrafish (Fig. 2E), suggesting the occurrence of retrograde blood flow from the ventricle back to the atrium. Additionally, no difference was observed in peak A (Supplementary Fig. 2C). However, the E peak varied widely, and in some instances, was obscured by the regurgitation wave or was altogether absent (Fig. 2E and Supplementary Fig. 2D). The phenotypic manifestations found in echocardiographs demonstrated that *tomato* zebrafish exhibited characteristics of valvular dysfunction.

### Cardiac morphological changes in *tomato* zebrafish

The outward appearance of the *tomato* zebrafish heart showed impressive enlargement. To determine the developmental origins of these cardiac abnormalities, the hearts of both WT and *tomato* zebrafish were extracted and compared at multiple developmental stages (1 mpf, 2 mpf, and 3 mpf). The most notable difference was atrial enlargement, evident as early as 1 mpf and sustained throughout subsequent developmental stages (Fig. 3A and Supplementary Fig. 3). As the fish matured, the atrium became so dilated that the extensive atrial tissue enveloped the ventricles in *tomato* zebrafish (Fig. 3A). The atrial myocardial tissue was nearly inelastic, and even a slight tension during dissection caused the heart tissue to rupture and bleed. The surface area of the overall heart (atrium + ventricle) and the ventricle were quantified at 3 mpf. The result showed that *tomato* fish had a twofold larger heart size compared with WT fish, while there was no significant difference in the size of the ventricle (Fig. 3B, C). These data indicated that the abnormal heart size in *tomato* zebrafish was mainly caused by atrial enlargement.

Masson's trichrome staining was performed to further investigate histological changes (Fig. 3D–I). An investigation was conducted on sections from different axes of both the WT and *tomato* zebrafish at 3 mpf. Section results from both the horizontal and coronal axes showed a consistent conclusion with the dissecting results (Fig. 3D–G), demonstrating significantly increased atrial size in the *tomato* line compared with the WT. In addition, hematoxylin and eosin (H&E) staining revealed disorganized and loosely arranged myocardial fibers in the *tomato* atria, indicating structural remodeling and potential functional impairment (Supplementary Fig. 4). No obvious morphological alterations were observed in the ventricular myocardium.

As previously mentioned, echocardiography revealed regurgitation of blood from the ventricle to the atrium. Cohort evidence has reported that the primary cause of blood regurgitation is the structural or functional

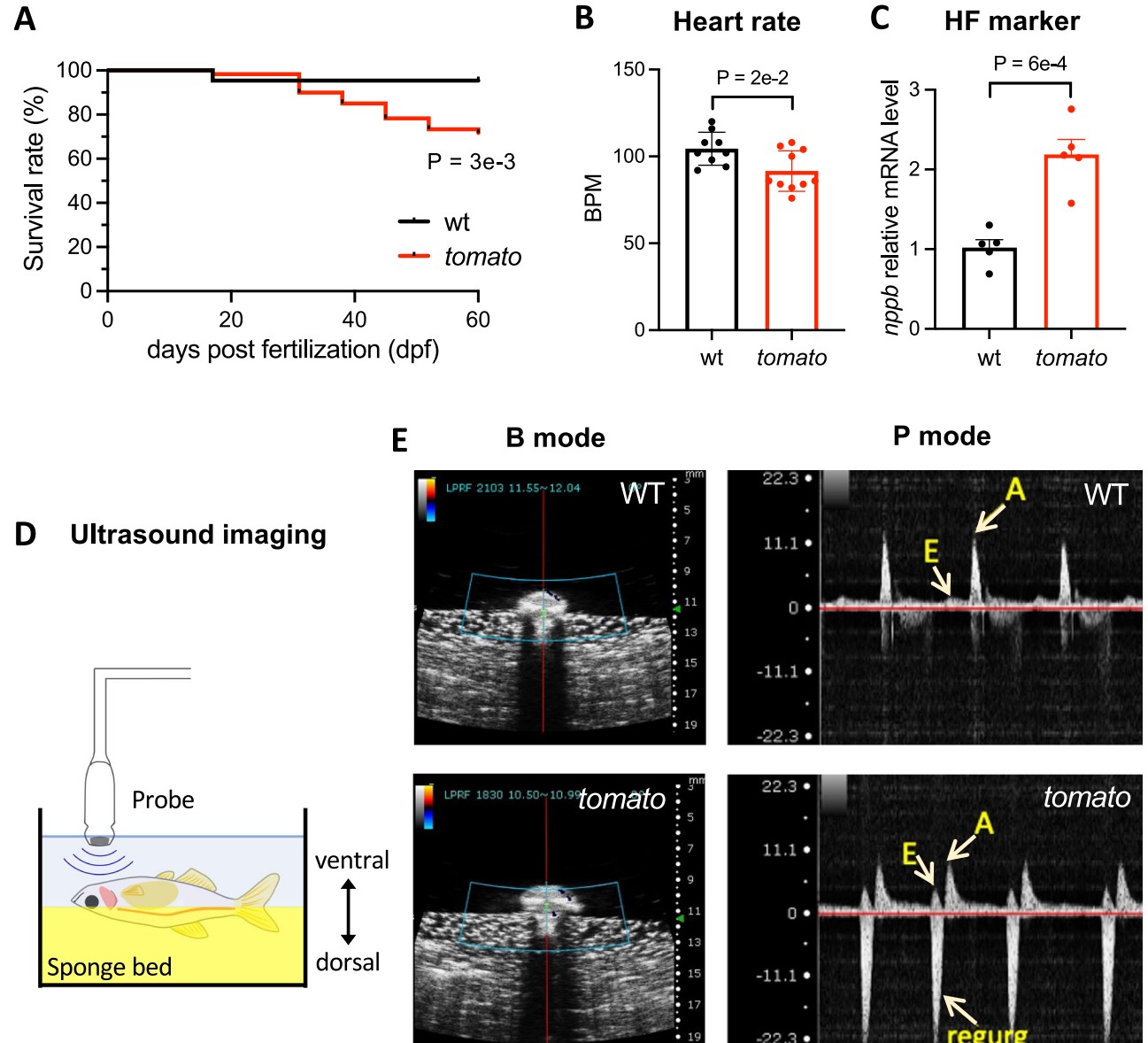

**Fig. 2 | *Tomato* zebrafish demonstrate apparent features of heart failure.**
**A** Survival rates recorded from 10 days post-fertilization (dpf) to 2 months post-fertilization (mpf) in wild-type (WT) (*N* = 22) and *tomato* (*N* = 60) zebrafish. **B** Heart rates measured in WT (*N* = 9) and *tomato* (*N* = 10) fish at 3 mpf. BPM beats per minute. **C** Relative expression of heart failure marker natriuretic peptide B (*nppb*) in hearts of WT and *tomato* zebrafish at 3 mpf. *N* = 5 fish in each group. Data were normalized to actin, beta 1 (*actb1*) expression. **D** Schematic of the experimental apparatus used for echocardiography. The anesthetized zebrafish are placed on a sponge bed ventrally upward, and the probe used to detect the short-axis view of the heart. **E** Representative pulsed-wave Doppler images of atrioventricular inflow in WT and *tomato* zebrafish. Brightness mode (B mode, left panel) represents the measurement location, while pulsed-wave Doppler (P-mode, right panel) shows the inflow pattern of each group. Arrows: early diastole wave (E), late diastole wave (A), and regurgitation (regurg). Y axis: velocity (cm/s). Data were presented as mean ± SEM. Statistics: Kaplan–Meier test in (**A**), unpaired two-tailed Student's *t*-test in (**B**, **C**).

impairment of the cardiac valve. Under high-magnification microscopy, a noticeable thickening of the AV valve between the ventricle and atrium was observed (Fig. 3H–J and Supplementary Fig. 5). Given that we suspected this valve thickening may impede the complete closure of the cardiac valve, leading to abnormal retrograde blood flow.

**Mechanisms of AV valve thickening in *tomato* zebrafish**
To gain further insight into the biological processes contributing to the morphological function of the AV valve, we conducted RNA sequencing of heart tissues from WT and *tomato* zebrafish at 3 mpf. Principal component analysis (PCA) revealed significant differences in gene expression between the WT and *tomato* (Fig. 4A). MA plot analysis was used to display the differences in expression patterns between

the WT and *tomato* groups, identifying 1208 upregulated genes (DESeq2, |fold change|>2, FDR <0.05) and 619 downregulated genes in *tomato* zebrafish (Fig. 4B; detailed list in Supplementary Data 2). Interestingly, gene ontology analysis using MSigDB revealed eight significant pathways, seven of which are shown in Fig. 4C (full list in Supplementary Table 1). Among them, six pathways were upregulated and two were downregulated (MSigDB Hallmark, |fold change|>2, FDR <0.05). There was a significant increase in the expression of genes associated with immune response pathways, including inflammatory response, tumor necrosis factor-alpha (TNFα) signaling via nuclear factor kappa-light-chain-enhancer of activated B (NFκB), and interleukin 2 (IL-2)-signal transducer and activator of transcription 5 (STAT5) signaling. Furthermore, EMT-related genes and markers of cell proliferation-

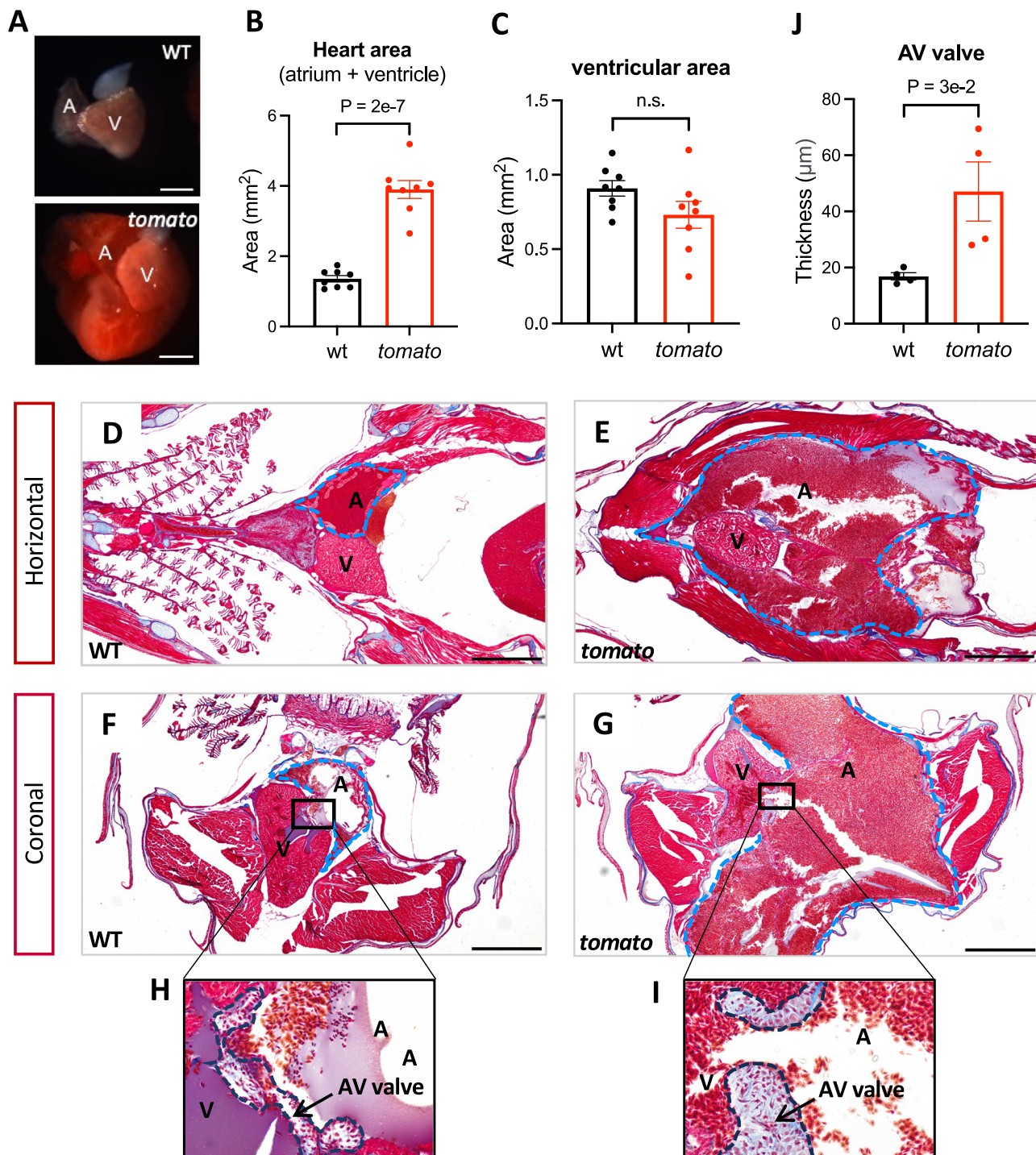

**Fig. 3 | Cardiac morphological changes in *tomato* fish. A** Representative images of hearts extracted from wild-type (WT) and *tomato* zebrafish at 3 months post-fertilization (mpf). Scale bars, 500 μm. **B** Heart areas (ventricle + atrium) of WT and *tomato* zebrafish. *N* = 8 fish in each group. **C** The area of the ventricle in WT and *tomato* zebrafish. *N* = 8 zebrafish in each group. **D–I** Masson's trichrome-stained sections. Horizontal (**D**, **E**) and coronal sections (**F**, **G**). Blue dotted lines: atrium. Scale bar, 500 μm. **H**, **I** Magnified images of AV valve in **F**, **G**, respectively. Dark blue dotted lines: atrioventricular (AV) valves. Scale bar, 50 μm. A atrium, V ventricle, arrow: AV valve. **J** Quantification of AV valve thickness. Three consecutive sections of each zebrafish were analyzed. The AV valve thickness was measured at its widest point in each section, and the average of these measurements was used as an indicator of valve thickness. Each dot represents a single zebrafish. *N* = 4 in each group. Data were presented as mean ± SEM. Statistics: unpaired two-tailed Student's *t*-test in (**B**, **C**, **J**).

related E2F targets, such as *mcm4, pcna, and trip13*, were significantly upregulated (Fig. 4D), suggesting increased proliferation and differentiation in the AV valve. Additionally, pathways related to myogenesis and fatty acid metabolism were markedly downregulated (Fig. 4C). To validate the RNA sequencing results, we performed qPCR analysis

targeting representative genes from key pathways. Inflammatory genes *marco* and *cybb* (Supplementary Fig. 6A), EMT-related marker *tfpi2*, *bmp6*, and *fossa* (Supplementary Fig. 6B), *sox8a* in IL-2–STAT5 pathway (Supplementary Fig. 6C), as well as TNFα–NFκB targets *nfkb2* (Supplementary Fig. 6D), all showed significant upregulation in *tomato*

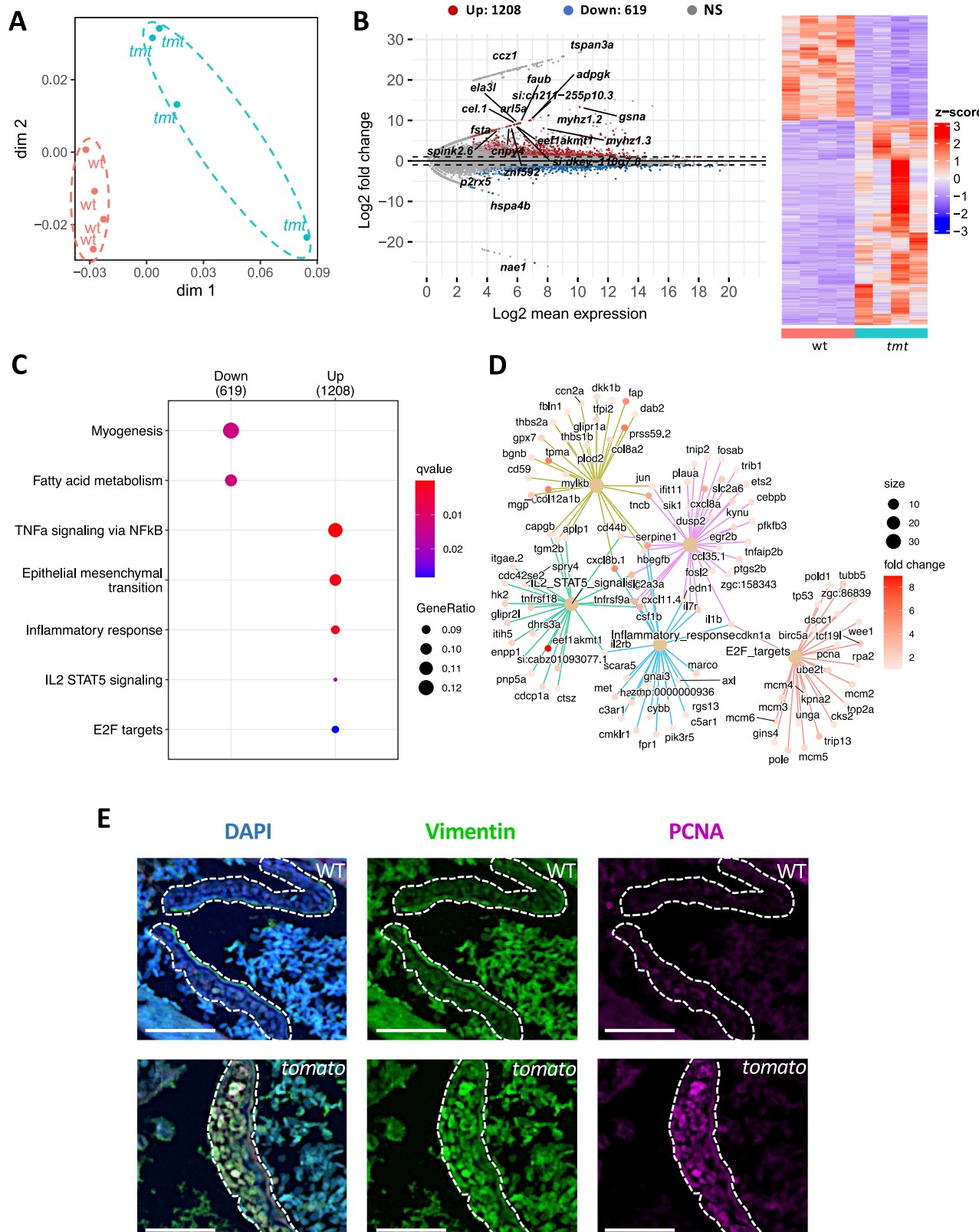

**Fig. 4 | Mechanisms of atrioventricular (AV) valve thickening in *tomato* fish.**
**A–D** RNA sequencing analysis of heart tissues from wild-type (WT) and *tomato* zebrafish at 3 months post-fertilization (mpf). *N* = 4 zebrafish in each group. **A** Principal component analysis of heart samples. **B** MA plot showed log2 (mean expression) and log2 (fold change) values for differentially expressed transcripts. Genes were upregulated in tomato hearts are colored red, while the genes that were downregulated in *tomato* hearts are colored blue. **C** MSigDB Hallmark enriched pathway showed the seven most significant differential pathways in *tomato* zebrafish hearts. **D** A gene network diagram based on the upregulated pathways. **E** Immunohistochemistry detection of the epithelial-mesenchymal transition (EMT) marker vimentin (green, middle panel) and the cell proliferation marker proliferating cell nuclear antigen (PCNA) (purple, right panel) in the AV valve. Merged images are shown in the left panel. 4',6-diamidino-2-phenylindole (DAPI) (blue) staining of cell nuclei. White dotted lines: atrioventricular (AV) valves. Scale bar, 50 μm.

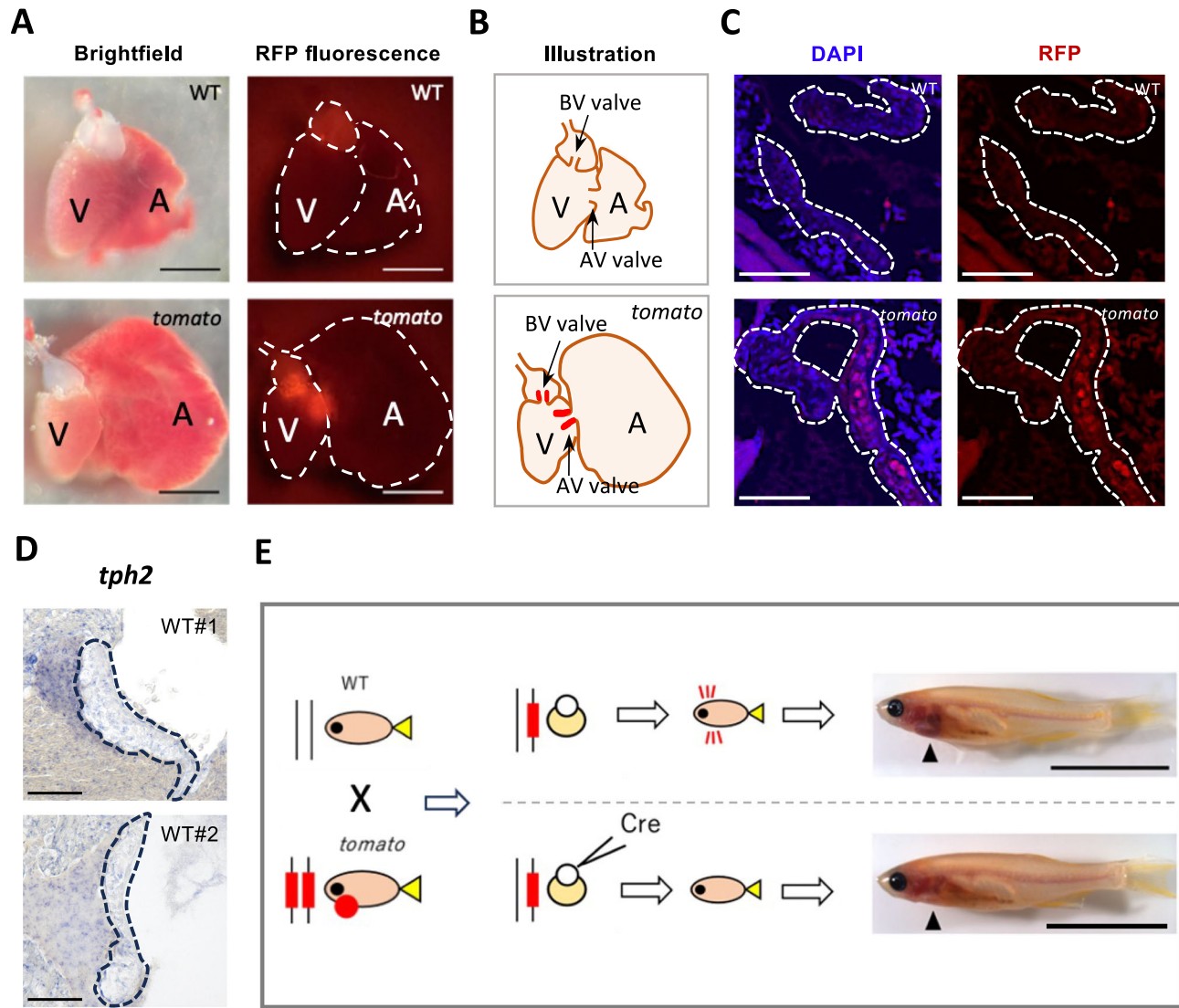

**Fig. 5 | Red fluorescent protein (RFP) accumulation and its implications in the atrioventricular (AV) valve of tomato zebrafish.** **A** Bright-field image (left) and RFP fluorescence (right) from WT and *tomato* zebrafish hearts. Scale bar, 1 mm. **B** Illustration of RFP expression (shown in red) in the AV and bulboventricular (BV) valves of *tomato* zebrafish. A atrium, V ventricle, AV valve atrioventricular valve, BV valve bulboventricular valve. **C** Immunohistochemical detection of RFP (red) expression in the AV valve region of WT and *tomato* zebrafish at 3 months post-fertilization (mpf). DAPI (blue) staining of cell nuclei. White dotted lines: atrioventricular (AV) valves. Scale bar, 50 μm. **D** *tph2* expression in the AV valve region of two WT zebrafish was detected by section in situ hybridization at 3 mpf. Black dotted lines: atrioventricular (AV) valves. Scale bar, 50 μm. **E** Schematic of Cre mRNA injection. Black arrows: heart position. Scale bar, 10 mm.

zebrafish. Furthermore, immunohistochemical analysis of the EMT marker vimentin and the cell proliferation marker PCNA was performed (Fig. 4E).

Increased staining intensity of both markers was observed in *tomato* hearts compared with WT, suggesting an overall elevation in EMT activity and proliferative potential in the AV valve region. Furthermore, investigation of fibrosis-related gene expression, including *mmp2*, *mmp9*, *mmp13a*, *col1a1a*, and *col1a2*, using qPCR (Supplementary Fig. 6E), revealed no significant changes between the groups.

In summary, these findings indicated substantial changes in EMT, cell proliferation and inflammation in the AV valves of *tomato* zebrafish compared to WT zebrafish, which was linked to significant pathological valve remodeling, while fibrosis appeared to potentially play a lesser role.

### DsRed accumulation and its implications in the AV valve of *tomato* zebrafish

As mentioned, the breeding experiment revealed that DsRed expression affected the phenotype of *tomato* zebrafish. Several studies have reported

that DsRed tends to aggregate and that the resulting cytotoxicity limits its biological applications[19]. We further speculated that the active EMT and cell proliferation processes were induced by DsRed expression and aggregation on the AV valve. To test this hypothesis, we first detected the location of DsRed expression using fluorescence microscopy. DsRed expression was detected in the heart region of *tomato* zebrafish at 14 dpf (Supplementary Movies 1–4), as the immature valve size in earlier development was difficult to observe. Similar patterns of DsRed expression were also observed in the dissected hearts of adult fish at 3 mpf (Fig. 5A), suggesting a persistent and long-term phenomenon. Two areas showed DsRed expression: the region connecting the atrium and ventricle, and the region connecting the ventricle and the bulbus arteriosus, implicating the involvement of the AV and BV valves (Fig. 5A, B). To further determine whether the red fluorescent protein DsRed was expressed in the AV valve of *tomato* zebrafish, tissue sections of 3 mpf zebrafish were analyzed by immunostaining using anti-RFP antibody. Compared with WT zebrafish, the valve interstitial cells (VICs) in *tomato* zebrafish were immunoreactive to anti-RFP antibody, confirming DsRed expression and accumulation (Fig. 5C).

To examine whether the DsRed expression is driven by the physiological expression of *tph2* in the valve, section in situ hybridization (ISH) was performed (Fig. 5D). Although *tph2* was expressed in the AV valve, the expression level was low and not noticeably different from that observed in surrounding cardiac and other tissues. Meanwhile, qPCR confirmed that *tph2* expression levels did not significantly differ between WT and *tomato* zebrafish (Supplementary Fig. 6F).

The transgene contained loxp sites to remove the DsRed-expressing genes (Supplementary Fig. 1A). To examine the effect of DsRed expression on the *tomato* phenotype, we injected Cre mRNA into embryos obtained from crossing WT and homozygote *tomato* and examined the phenotype of the injected fish (Fig. 5E). Without Cre mRNA injection, the embryos were DsRed-positive and later developed heart enlargement. Cre mRNA-injected embryos exhibited no red fluorescence (39/40). In addition, adult fish growing up from the injected embryos showed no obvious heart defects. Despite the removal of the DsRed gene, the Cre mRNA-injected fish still harbored yellow fluorescent protein (*yfp*). These results demonstrated that DsRed was essential for the development of the heart defect phenotype in *tomato* zebrafish.

## Exploration of the transgene insertion site via whole genome sequencing (WGS)

To investigate the genomic integration sites of the exogenous transgene, WGS was performed on the *tomato* DNA sample. Sequence reads were mapped to both the zebrafish reference genome (GRCz11) and the full-length plasmid sequence (Supplementary File 1), with high-homology regions masked to minimize false positives. The relative read depth between host and vector sequences at these regions ranged from 6X to 10X, suggesting the presence of multiple transgene copies. Initial SV analysis using automated software pipelines, Dynamic Read Analysis for GENomics (DRAGEN; Illumina), did not identify any statistically significant breakpoints indicating transgene insertion because the discordant and split reads were mapped to multiple loci. Supplementary manual inspection with the Integrative Genomic Viewer (IGV)[20] identified four discordant and twelve split reads, which all shared a highly identical sequence segment (Supplementary Fig. 7). Based on this sequence, combined with PCR, Sanger sequencing, and BLASTN analysis, we identified an ~200 bp genomic fragment that is presumably adjacent to the transgene (Supplementary File 2). However, owing to its multimapping nature, the precise genomic insertion site of the transgene remains unresolved.

## Discussion

In the present study, we demonstrated that the novel transgenic zebrafish *tomato* displayed chronic valvular heart disease and heart failure. The transgenic line manifested several characteristics of valvular diseases, presenting both morphological and functional abnormalities of the AV valve. In addition, the elevated mortality rate, reduced heart rate, and higher *nppb* expression level in *tomato* suggested the presence of heart failure. In *tomato* fish, evidence of DsRed presence in the valve was detected as early as 14 dpf, which preceded any other cardiac abnormalities. Cre-loxP-mediated removal of DsRed successfully rescued the phenotype, implicating DsRed as a key driver in the development of the cardiac defect. Collectively, we reported a DsRed-induced, progressive model of cardiac dysfunction that manifests characteristics of valvular dysfunction and heart failure.

In this study, we elucidated the in vivo effects of DsRed aggregation in zebrafish heart tissues. Consistent with previous findings, upregulated cell proliferation and EMT contributed to valve thickening in *tomato* zebrafish[21]. Except that, RNA-seq analysis revealed the upregulation of inflammatory responses in *tomato* zebrafish hearts. Numerous studies have demonstrated that the stress response induced by protein aggregation is mitigated through inflammatory responses[22]. Simultaneously, prolonged infiltration of immune cells and cytokines stimulates cell proliferation and differentiation[23]. This complete pathway explains how DsRed aggregation leads to valve thickening. The affected valve, with its limited mobility,

further restricts blood flow and regurgitation, ultimately leading to blood retention in the atria and atrial enlargement. Associated reports commonly observe fibrosis after tissue injury, an effect that is irreversible and causes disruption of tissue architecture[24]. Thus, fibrosis is an inevitable in the mechanism of valve thickening. However, contrary to expectations, fibrosis did not contribute to valve thickening in this model, unlike previous studies in mouse models[21,25]. This discrepancy may be attributed to the unique cardiac regeneration capability of zebrafish, where fibrosis is transient and its regression is concomitant with the regeneration of normal tissue.

DsRed is a commonly used imaging application in biological research, allowing for the detection of the expression and localization of relevant molecules in tissues[26]. However, its tendency to oligomerize and aggregate into cytotoxic higher-order forms is widely considered a disadvantage, limiting its broader application[27,28]. In the present study, we demonstrated a unique heart failure model caused by DsRed aggregation in valve cells, revealing more possibilities for its application. By properly applying the characteristics of DsRed, its drawbacks can also be converted into distinct advantages. Previous studies have reported that DsRed aggregation causes proteinopathy in cardiomyocytes in a transgenic mouse model, thus demonstrating the feasibility of using DsRed's characteristics[25].

The accumulation of abnormal proteins and other substances in the cardiac valve leads to valvular heart disease. With the continuous development and improvement of diagnostic methods, relevant cases have been reported. For example, amyloidosis in heart tissue, traditionally associated with cardiomyocytes, is now frequently reported in the aortic, tricuspid, and mitral valves[29]. Besides, glycosaminoglycan (GAG) aggregates have been detected on the mitral valves of patients with mucopolysaccharidoses[30]. The pathological consequences caused by the aggregation of proteins or GAGs are similar to those observed for DsRed aggregation on the AV valve in our experiments, which led to progressive valve thickening, valve functional impairment, blood regurgitation, atrial enlargement, and diastolic dysfunction. The severity of valvular heart disease with this mechanism is extremely high, ultimately leading to heart failure and posing a serious threat to patients' lives. The conventional treatment is valve repair or replacement surgery, but the prognosis varies among individuals[31–33]. Based on similar mechanisms of occurrence and progressive pathogenesis, our model serves as a valuable tool for studying the subsequent processes of protein accumulation and exploring potential treatment methods.

Since the transgenic component was designed to express DsRed under the control of the *tph2* promoter, we first predicted that DsRed expression would be driven by the physiological expression of *tph2* in the valve. Contrary to expectations, in situ hybridization revealed a low-level expression of *tph2* in the AV valve, similar to background levels. This result suggests that the presence of DsRed cannot be solely attributed to the physiological expression of *tph2* in the valve, implying other valve-specific mechanisms that contribute to the phenotype.

Despite performing WGS, we were unable to identify the definitive insertion site for the transgene. This is likely due to limitations inherent to short-read sequencing. However, crossbreeding revealed that the inheritance pattern of DsRed followed Mendelian laws, supporting our hypothesis that a single insertion locus exists. In combination with the relative read depth, the most plausible explanation is that this locus harbors multiple (~6–10) tandem copies of the transgene. This interpretation is further strengthened by Cre recombination experiments, where removal of the DsRed segment while retaining the remaining ChR2-YFP sequence still rescued heart defects. These findings suggest that the cardiac enlargement phenotype is not attributed to insertional disruption of endogenous genes, but is rather associated with DsRed expression itself. Further investigations using long-read sequencing and targeted functional studies will be required to determine the precise insertion site and clarify the role of DsRed in cardiac valve pathology in *tomato*.

As a model of heart failure, zebrafish has many advantages over other animal models. For instance, regarding cardiac physiology, zebrafish heart rate is similar to that of humans, making it more comparable and referable in heart rate variability[34]. Additionally, despite possessing only one atrium and

one ventricle, zebrafish have more similar action potential dynamics and electrocardiographic profiles to humans compared with other animal models, enabling a simple and precise simulation of human cardiac physiology[35]. Zebrafish have also been utilized in research on cardiac valves[10]; however, most research has heavily focused on the developmental processes[8,36] and phenotypes present in embryonic to early larval stages[37,38]. Our current model exhibited no overt apparent cardiac abnormality during the embryonic stages (approximately 72 hpf); however, a progressive chronic cardiac phenotype emerged following the appearance of DsRed in the AV valve at 14 dpf. At 30 dpf, cardiomegaly became prominent, reflecting the atrial enlargement. Heart enlargement progressed by 3 mpf, at which point AV valve thickening, regurgitation, and an elevated level of *nppb* expression were confirmed. This series of cardiac phenotypes suggests a possible pathology of the model: first, DsRed accumulates in the AV valve, leading to the valve's thickening and insufficiency. This valvular dysfunction then causes atrial extension by excessive blood volume and possibly heart failure. The progressive cardiac phenotype of *tomato* is distinct from previously reported cardiac disease models in zebrafish that show signs of heart failure[14,39,40] or AV valve dysfunction[37,38] during the early stages of life. In addition, the increase in mortality rate in *tomato* is lower than in the existing models. Many adult *tomato* fish survive for >1 year, which is comparable to the WT. This extended lifespan makes them ideal models for investigating aging heart failure. Moreover, *tomato* can easily be distinguished from its WT littermates by simply observing DsRed fluorescence in the dorsal raphe nucleus. This non-invasive identification method allows transgenic line detection as early as 3 dpf. Therefore, it enables researchers to observe the progression of heart failure and valvular lesions from early larval stages. Taken together, the *tomato* could serve as a powerful cardiac disease model that mimics an aspect of human heart diseases that was not illustrated by the existing model. Adult zebrafish heart failure models can also serve as valuable tools for long-term drug screening, cardiac toxicity assessment, and cardiac regeneration research.

The mechanism through which valvular defects contribute to heart failure in zebrafish remains unclear. However, given that valvular diseases such as tricuspid or mitral regurgitation are well known to contribute to heart failure in humans[41,42], investigating this relationship in zebrafish remains biologically relevant. We acknowledge that our study provides limited functional evidence; however, the cardiac phenotypes observed in the *tomato* line suggest that it may reflect certain aspects of the valvular disease-induced heart failure. In addition, the methodology to quantify cardiac function in juvenile and adult zebrafish has not been established, except for performing echocardiography, requiring specific equipment[43]. Although these knowledge gaps and technical restraints remain a possible limitation in the present study, we expect that our transgenic line could serve as a model to further study zebrafish valvular dysfunction and heart failure in their adulthood.

In summary, our study introduced a novel zebrafish model of valvular heart disease and heart failure characterized by gradual heart enlargement and blood regurgitation, which mimics human heart failure progression. We elucidated the mechanism underlying DsRed aggregate-induced valve thickening, highlighting the role of the EMT as well as cell proliferation and inflammation, while noting the unique aspect of DsRed aggregate cytotoxicity in establishing disease models. This model provides valuable insights into the pathogenesis of aging heart failure and may serve as a platform for future drug screening and therapeutic development.

## Methods
### Zebrafish husbandry
All zebrafish husbandry procedures were performed under standard conditions in accordance with Kyoto University's ethical and animal welfare regulations and were approved by the Animal Care Committee of Kyoto University (approval number: Med Kyo 21006). Wild-type and transgenic zebrafish used in this study were derived from the transparent *casper* double mutant background (*mitfa*$^{w2/w2}$; *mpv17*$^{a9/a9}$)[44]. Zebrafish were maintained in

a circulation system at 28 °C with a light: dark cycle of 14:10 h[45]. The adult experiments were conducted on 3 mpf zebrafish, while the ages of the juvenile zebrafish are specified in the text.

### Generation of the transgenic fish line
Tol2-based transgenesis was performed to generate a new transgenic zebrafish line, *Tg(-4.9tph2:LOXP-DsRed express-LOXP-ChR2-YFP); casper (mitfa*$^{w2/w2}$; *mpv17* $^{a9/a9}$) as described previously[46]. Approximately 25 ng of the plasmid carrying the artificial gene, along with 25 ng of *Tol2* mRNA, were injected into one-cell stage of *casper* embryos. The injected embryos were raised and maintained in a heterozygous background by crossing with *casper (mitfa*$^{w2/w2}$; *mpv17* $^{a9/a9}$). Obtained fish were identified by observing the RFP fluorescence in the zebrafish brain at 3 dpf using an Olympus SZX16 stereomicroscope (OLYMPUS). Zebrafish displaying RFP fluorescence were categorized into the *tomato* group, whereas those lacking RFP fluorescence were classified into the WT group. Supplementary File 1 shows the complete sequence of the plasmid used for the generation of the transgenic line, and the structure of the transgene is shown in Supplementary Fig. 1A.

### Cre-mRNA injection
Embryos were obtained by breeding *tomato* homozygous fish crossed with *casper (mitfa*$^{w2/w2}$; *mpv17* $^{a9/a9}$) fish. Cre mRNA was synthesized from pCS2+ Cre as described previously[47]. Approximately 1 nl of Cre mRNA (50 ng/ml) was injected into the one-cell stage of the obtained *tomato* embryos. After injection, red fluorescence was observed and recorded at 3 dpf.

### Survival and heart rate analysis
After assessing the DsRed expression, DsRed-negative wild-type (WT) fish and DsRed-positive *tomato* were placed into separate tanks. All groups were maintained under the same density and feeding conditions. From 10 dpf, the number of surviving fish and frequency of mortality events were recorded weekly.

To assess heart rate, zebrafish were anesthetized in 0.016% tricaine (#21438-82, Nacalai Tesque) for 2 min. This approach ensured minimal effects on adult zebrafish heart rate and cardio-depression. Anesthetized zebrafish were carefully transferred to a petri dish and positioned under a stereomicroscope for visualization and recording. Heart rates were measured once the heart rhythm had stabilized, with a 30 s recording. Following the completion of heart rate recording, the zebrafish were promptly revived in a recovery tank filled with fresh system water to ensure a rapid return to baseline physiological states. Nine WT fish and 10 *tomato* fish were analyzed.

### Doppler echocardiography
Zebrafish were anesthetized in 0.016% tricaine solution diluted in system water for 1 min. The anesthetized fish were then transferred to a sponge bed and placed in the ventral position, facing upward into a groove manually cut in the center of the sponge bed. The same concentration of tricaine solution was added to the container with a sponge bed until the zebrafish were entirely submerged to ensure that they remained still during the detection process at room temperature. Imaging was performed using an ultrasound system (Prospect T1, S-Sharp Corporation) equipped with a 40 MHz probe. The probe was held perpendicular to the horizontal plane with the probe head positioned below the water surface. A short-axis view was obtained. Brightness mode (B-mode) imaging was used to visualize the coronal plane of the heart. The depth and position of the probe were adjusted to display the atrioventricular valve. Pulsed-wave Doppler (P-mode) was used, with the Doppler gate placed on the atrioventricular valve area to detect ventricular inflow. The flow data were recorded for 30 s after imaging stabilization. The entire process was completed within 3 min. Following completion of the imaging procedure, the fish were transferred to a tank with fresh system water for recovery. No fatalities occurred during the process. Data analysis was performed using Prospect T1 software. Cardiac function was evaluated by quantifying the velocities of the E and A peaks as well as the E/A ratios.

## Histological and immunohistochemical analysis

For histological analysis, three fish were collected from the WT and *tomato* fish at 3 mpf. The zebrafish were quickly euthanized in cold 1x phosphate-buffered saline (#314-90185, Nippon Gene Material) at pH 7.4. After euthanasia, each fish was dissected from the anal fin to the tail fin using a razor blade and the tail parts were discarded. The head parts were fixed in 4% paraformaldehyde (#09154-56, Nacalai Tesque) overnight on a rocking shaker at 4 °C. The following morning, the samples were sent to the Anatomy Center of Kyoto University for paraffin embedding and sectioning in three planes: coronal, horizontal, and sagittal. Hematoxylin and eosin and Masson's trichrome staining were performed on the paraffin-embedded sections.

Immunohistochemistry was performed at the Anatomy Center of Kyoto University. Three samples were collected from each group and sliced along the coronal plane. For RFP staining, sections were immunostained with an RFP monoclonal antibody (1:500, MA5-15257, Invitrogen) as the primary antibody and goat anti-mouse IgG (H + L) Alexa Fluor Plus 647 (1:500, Invitrogen) as the secondary antibody. Vimentin staining involved the pretreatment of sections with citrate buffer for 10 min in a microwave, followed by immunostaining with an anti-vimentin polyclonal antibody (1:100, GTX133061, GeneTex) and goat anti-rabbit IgG (H + L) Alexa Fluor Plus 488 (1:500, Invitrogen) as secondary antibody. For proliferating cell nuclear antigen (PCNA) staining, the sections were treated with citrate buffer for 20 min in a microwave. Immunostaining was performed using anti-PCNA monoclonal antibody (1:500, P8825, Sigma-Aldrich), followed by incubation with goat anti-mouse IgG (H + L) Alexa Fluor Plus 647 (1:500, Invitrogen). All sections were imaged using the all-in-one fluorescence microscope BZ-X800 (Keyence) and the BZ-X800 viewer software.

## Whole-mount and section in situ hybridization

For whole-mount in situ hybridization, 3 dpf WT zebrafish were fixed with 4% paraformaldehyde (PFA). For section in situ hybridization, 3 mpf fish heads and thoraxes were fixed with 4% PFA, followed by demineralization with EDTA-based solution G-chelate mild (GCM-1, Genostaff) for 10 days. The fish were embedded in paraffin and sectioned. In situ hybridization was conducted essentially as previously described. Briefly, *tph2* cDNA corresponding to 498–1051 nt was amplified and cloned into pBluescript II KS. Digoxigenin-labeled RNA probes were synthesized and hybridized at 56 °C for 16 h, followed by immunodetection using Anti-Digoxigenin-AP, Fab fragments (11093274910, Roche). For color development, 4-nitro blue tetrazolium chloride (NBT) and 5-bromo-4-chloro-3-indolyl-phosphate (BCIP) were used. Imaging was performed using a stereomicroscope (SZ61, OLYMPUS) for whole-mount in situ hybridization and an upright microscope (BX43, OLYMPUS) for section in situ hybridization.

## Genomic DNA extraction for WGS

Genomic DNA was extracted from adult *tomato* zebrafish liver tissue using DNeasy Blood & Tissue Kit (#69504, Qiagen) following the manufacturer's protocol. Briefly, tissue was homogenized in ATL buffer using Biomasher II (Nippi), followed by proteinase K treatment at 56 °C for 1 h until complete lysis. RNA was removed by RNase A (#19101, Qiagen) treatment. The lysate was then applied to a spin-column, washed with AW1 and AW2 buffers, and eluted in AE buffer. DNA concentration and purity (ratio of the absorbance at 260 and 280 nm ($A_{260/280}$) and $A_{260/230}$) were assessed using a NanoDrop One (Thermo Fisher). The DNA sample selected for WGS had a concentration of >80 ng/µl in 50 µl, and met quality thresholds of $A_{260/280} \geq 1.6$ and $A_{260/230} \geq 1.6$.

## WGS and subsequent analysis

Short-read WGS was performed by Takara Bio Inc. Genomic DNA (2.2 µg) was fragmented using Covaris to obtain DNA fragments of several hundred base pairs. Genomic DNA libraries were prepared using the TruSeq DNA RCP-Free Library Prep Kit (Illumina) and indexed with IDT for Illumina—TruSeq DNA UD Indexes v2 (Illumina), following the

manufacturer's protocol. The quality and fragment size distribution of the final libraries were evaluated using the TapeStation HSD5000 assay (Agilent Technologies).

Sequencing reads were mapped to both the zebrafish reference genome (GRCz11) and the full-length transgene vector sequence using DRAGEN Bio-IT Platform (v4.3.6). To minimize false-positive structural variants (SV), highly similar sequence regions between the genome and transgene were masked before analysis using BLAST (v2.13.0+) and bedtools (maskFastaFromBed, v2.28.0). For SV detection, discordant read pairs and split reads were extracted from alignment files using SAMtools (v1.10) and LUMPY (v0.2.13). Manual inspection of candidate insertion signals was conducted using the Integrative Genomics Viewer (IGV). SV annotation was performed with SnpEff (v4.2) to assess potential functional impacts of the identified breakpoint.

## RNA extraction and quality assessment

The extracted hearts were dissected and kept in RNAlater (#AM7021, ThermoFisher) solution. Five samples were prepared for WT and *tomato*, and two hearts were pooled per sample. Total RNA was extracted from the cell suspension using RNeasy Fibrous Tissue Mini Kit (#74704, Qiagen) following homogenization in buffer RLT. RNA quality was assessed using RNA ScreenTape (#5067-5576, Agilent Technologies), with the TapeStation 4150 system (Agilent Technologies). RNA samples with an RNA Integrity Number Equivalent (RINe) >8 were used for further analyses. The RNA samples were stored at −80 °C until use.

## Quantitative PCR

Total RNA was reverse transcribed into complementary DNA using the High-Capacity RNA-to-cDNA kit (#4387406, ThermoFisher). qPCR was performed using Applied Biosystems QuantStudio 6 Flex and QuantStudio 3 (Applied Biosystems) with SsoAdvance Universal SYBR Green Supermix (#1725271, Bio-Rad). In this research, the target genes from PrimePCR™ SYBR Green Assays (Bio-Rad) were: matrix metalloproteinase-2 (*mmp2*) (qDreCID0004788), *mmp9* (qDreCID0002041), *mmp13a* (qDreCED0018863), collagen, type I, alpha 1a (*col1a1*) (qDreCED0017616), *col1a2* (qDreCED0019386), *nppb* (forward 5' TGTTTCGGGAGCAAACTGGA 3' and reverse 5' GTTCTTCTTGGGACCTGAGCG 3'), *marco* (qDreCED0010322), *cybb* (qDreCED0014759), *il7r* (qDreCID0005046), *tfpi2* (qDreCID0002865), *bmp6* (qDreCID0014051), *fosas* (qDreCED0015103), *cd44a* (qDreCID0001835), *nfkb2* (qDreCID0020554), *tbx1* (qDreCED0015105), *sox8a* (forward 5' AAACTCGCCGATCAGTACCC 3' and reverse 5' TGCAGCCTCAGTCTTTCAGC 3'), *tph2* (qDreCED0017433) and the housekeeping gene *actb1* (qDreCED0020462). The relative gene expression levels were analyzed using the $2^{-\Delta\Delta CT}$ method. The delta (ΔΔCt) values were normalized to the ΔΔCt of *actb1*. Each sample was analyzed in either triplicate or duplicate.

## RNA sequencing analysis

Total RNA were extracted from the heart tissues of 3 mpf *tomato* fish and their WT siblings. Two hearts were pooled into a single sample. Both the WT and *tomato* samples were prepared as four replicates. Libraries for RNA sequencing were prepared by Macrogen Japan Co. as previously described[45]. A total of 200 ng of RNA was used for library preparation using the TruSeq Stranded mRNA LT Sample Prep Kit Set A (Illumina). The sequencing of the library was performed using the NovaSeq 6000 sequencing system (Illumina) with paired-end 101-bp reads.

The raw sequencing reads were subsequently subjected to quality control analysis, adapter trimming, and low complexity filtering using fastp[48]. The reads were then mapped to the *Danio rerio* GRCz11 reference genome using Hisat2[49]. StringTie was used to assemble and quantify the alignments in each sample[49]. The visual sequencing results were analyzed on the RNAseqChef website (https://imeg-ku.shinyapps.io/RNAseqChef/)[50]. Pair-wise differential gene expression analysis was performed using DESeq2 with the following criteria: fold change ≥2 and false discovery rate (FDR)

≤0.05. MA plots and PCA were automatically generated online. The enrichment analysis was performed to cluster the set of genes based on the MSigDB Hallmark gene set and determine their statistical significance.

## Statistical analysis

All graphs and statistical analyses were generated using GraphPad Prism version 10.1.1 (GraphPad Software, Inc.). The unpaired two-tailed Student's *t*-test was utilized to compare differences between the two groups. The Kaplan–Meier test was used to assess the survival rates. All quantitative results were presented as means ± standard error of the mean (SEM). The sample size (*N*) represents the number of animals. Statistical significance was set at $P < 0.05$.

## Reporting summary

Further information on research design is available in the Nature Portfolio Reporting Summary linked to this article.

## Data availability

The WGS and RNA-seq data generated in this study have been deposited in the NCBI Sequence Read Archive (SRA) PRJNA1310275. The source data for all statistical graphs are provided in Supplementary Data 1. Additional information is available from the corresponding author upon reasonable request.

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

## Acknowledgements
This work was supported by the Japan Society for the Promotion of Science (JSPS) (KAKENHI grant numbers 21H05042, 21H05326, 22H00986, and 23H03883) and JST SPRING (Grant Number JPMJSP2110). The authors are grateful to the Center for Anatomical, Pathological and Forensic Medical Research, Kyoto University Graduate School of Medicine, for technical assistance with the histopathological preparation. We are also grateful to Yuta Aoyama for helping to maintain the zebrafish. We acknowledge the support of the NEPA Gene in lending us the ultrasound imaging system. We also thank the Center for Medical Research and Education, Graduate School of Medicine, the University of Osaka, for their support.

## Author contributions
S.M. established the zebrafish model. S.M. and R.K. together conceived the project. Y.L. and S.M. conducted the experiments and analyzed the data. S.R.S. and T.N. helped with technical support. Y.L. and S.M. wrote the first version of the manuscript. R.K. and M.H. supervised the project. S.M., R.K., S.R.S., T.N., and M.H. reviewed and edited the manuscript. All authors read and approved the final manuscript.

## Competing interests
The authors declare no competing interests.
