## [Transparent Peer Review file · Communications Biology]

A zebrafish model of chronic heart failure caused by protein aggregation in heart valves

Corresponding Author: Professor Ryo kimura

Version 0:

Reviewer comments:

Reviewer #1

(Remarks to the Author)

Heart failure (HF) is a major clinical problem worldwide, and current treatments do not address the underlying cause and are mainly symptomatic, hence further research is needed. One major problem in this area of research is the lack of appropriate research models. Multiple factors, including cardiac dysfunction, high blood pressure, and genetic abnormalities, chronically cause HF, making it difficult to recapitulate in an animal model. Zebrafish have recently been used to model human heart disease, mainly inherited diseases caused by genetic alterations. It could be useful to generate an experimental model that recapitulates phenotypes observed in patients with HF. The authors identified a new TG zebrafish line which develops late cardiac phenotypes; this includes an enlarged atrium and AV valve abnormalities. Their RNA-seq analysis suggested potential candidate genes involved in these phenotypes. Although their TG zebrafish has potential as a human heart disease model, there is not enough data to support their claims.

Major issues:

There is little characterisation of the TG (tomato) used, particularly its genetic background, making the usefulness of these animals as models of heart failure unclear. One of the critical points is to determine the locus where the transgene is inserted and to assess whether this insertion affects levels of gene expression at that locus. The authors showed that removal of DsRed and BGH-pA by Cre recombination rescued the phenotypes. However, the other transgenes, including Tol2 element, tph2 promoter, Chr2, YFP, SV40-pA, remain, so it is unclear why only DsRed and BGH-pA lead to cardiac phenotypes and other transgenes do not.

The authors mentioned phenotypes in heterozygotes without data that need to be shown in figures. The phenotypic penetrance in the homozygous TG should be determined. They argue that RFP expression in the TG is observed in serotonergic neurons. However, there are no experiments to determine which cell types express RFP by immunostaining or using a reporter fish.

They found defects in the atrium and AV valve in the TG hearts and no significant changes in the size of the ventricle. It would be worth to analyse EF and FS in the ventricles of the TG. Given that patients with HF have defects in the ventricles, such as reduced cardiac function, and morphological defects, it is thus not clear whether their TG fish model can be useful to study human HF due to minimal data provided.

The authors show cardiac phenotypes only in adult TG, but it is also important to investigate phenotypes in embryos and determine when heart size, cardiac function, AV morphology, and nppb levels are affected.

They performed RNA-seq analysis in 3 mpf fish, which have already developed phenotypes, making it difficult to study the direct effects of the plasmid insertion leading to AV phenotypes.

Minor issues:

The name of the TG needs to be in accordance with the rules of ZFIN (Zfin.org) so that people in the field can obtain much information about the type of TG.

Using 'Tomato', 'RFP', and 'DsRed Express' interchangeably may lead to confusion.

In Figure S1 A, non-English language is used.

Reviewer #2

(Remarks to the Author)

The authors introduce a model of chronic heart failure, generated by the introduction of a *thp2:DsRed* construct. The data presented are interesting (including the RNASeq data, and the adult doppler data). While this could be a very valuable model to study long-term effects of retrograde flow in the adult heart, the model needs to be further validated and explained.

Major concerns.

- 1) It is unclear how many insertions of the construct are in the "tomato" mutant
- 2) The authors make a rescue experiment with injecting Cre mRNA in heterozygous fish. They briefly describe that tomato heterozygous embryos as showing "an expanded heart" but do not present or analyze further the heterozygous phenotype and how similar it is to the homozygous phenotype. It seems that the genetics of the heart failure phenotype are recessive (page 11, figure 2A), which makes it unlikely to be an RFP protein aggregation phenotype (that would be dominant) and points to a locus specific disruption of a gene. The genomic context of the insertion(s) should be identified by flanking DNA sequencing and presented.
- 3) It is unclear if the DsRed protein they see in the adult valves is driven by the physiological expression of the *thp2* gene. An in situ should at these stage should be performed to show if this kind of neurons exist in the valves or elsewhere in the adult heart.
- 4) Overall the immunohistochemistry pictures are of low resolution and the heart failure phenotype not very well documented and supported. The atrium looks indeed dilated but what about the cardiomyocyte phenotype? Is there an atrial specific hyperplastic response or are the atrial cardiomyocytes bigger? And what about the ventricular phenotype (ventricles appears to be smaller in the mutants, is there apoptosis or any other explanation for this phenotype?) . Higher magnification figures of adult cardiomyocyte size/shape in atrium and ventricle should be included.
- 5) The valves presented in Figure 3H-I do not appear to be at comparable positions, which admittedly the atrial dilation makes it very tough. Could they make a compilation of the other 3 fish they measured (wt and tomato) as a supplementary figure, so one can appreciate the differences in thickness.

Reviewer #3

(Remarks to the Author)

Summary:

In this manuscript, Li et al. reported a novel model of heart failure zebrafish caused by protein aggregation in the cardiac valve. They introduce a progressive transgenic zebrafish model that modulates red fluorescent protein (RFP) expression under the *thp2* promoter and displayed severe heart enlargement during the larval stage, which increased with age. Further cardiovascular phenotypic diagnosis revealed that this zebrafish model was associated with symptoms consistent with heart failure. These findings will help to advance our understanding of the effects of TPH2 on heart valves and highlight the potential of this transgenic Zebrafish to be used to investigate chronic heart failure and develop new therapies. Their study provides new insights into valvular disease-induced heart failure and offers new ideas for future research on valvular function and its potential progression to heart failure. Overall, the study is highly significant in the field of heart failure research. They carried out the experiments in methodical way. The results are presented structurally well. Despite its high significance, the study needs substantial improvements. Please see the comments below,

Comments:

1. In the result, the authors states that no distinct phenotypes were observed in the mutants during the embryonic phase. However, as tomato fish aged, they unexpectedly developed expanded hearts. On contrary, in the introduction, they have mentioned that severe heart enlargement was observed during the larval stage, which increased with age. Please clarify.
2. Did the authors perform heart rate analysis in the developmental stages? They should present this analysis to support their claim.
3. The validation of gene expression for at least the highlighted genes in each pathway for example the inflammation etc. will be necessary to confirm the finding.
4. In the supplemental video files the heartbeat in mutant appeared faster than the wt. please check.
5. A brief information about the methodology used to construction of the transgenic will be helpful in reproducibility.
6. The authors mentioned that they measured the weight and length of each fish analyzed. further they also mention about the genders. However, there are no results presented related to this analysis. What is the significance of these variables?
7. The authors should analyze the expression of marker genes for the AV valve development to support the AV thickness observation.
8. Identification of the early affected set of genes be helpful to uncover the regulatory role in the onset of the heart enlargement phenotype.
9. An expression analysis of *thp2* to investigate its expression in the heart valve will be helpful. In-situ hybridization would be a feasible approach.
10. Does the sequencing analysis reveal expression of *thp2* in the mutant? It will be interesting to see how the expression of *thp2* or related genes changes in mutant.
11. To assess heart rate, zebrafish were anesthetized in 0.2 mg/ μ L tricaine, while Zebrafish were anesthetized in 0.016% tricaine solution diluted in system water for 1 min for ultrasound. Why the doses of tricaine different in these two methods in the same study?

Reviewer comments:

Reviewer #1

(Remarks to the Author)

The authors have addressed all questions, so there are no more concern.

Reviewer #3

(Remarks to the Author)

The revised manuscript has improved significantly. Overall, the results presented in the manuscript align with the conclusions as stated by the authors. Moreover, the authors have addressed the majority of the comments and improved their manuscript extensively. I believe that the work is in the scope of the journal and significant to the field. I have no further comment but a few suggestions to add as discussed below. Please discuss the following in the discussion wherever applicable.

1. The presence of multiple insertions of a construct (e.g., six copies as in your tomato zebrafish mutant) can significantly impact the interpretation of the mutant phenotype and its linkage to specific genes. For instance, each insertion can disrupt different genomic loci, potentially affecting the expression or function of multiple genes. This complicates the attribution of phenotype to a single insertion or gene. The observed phenotype may be a composite of several disruptions. Please discuss this in your discussion.

2. I believe incorporating additional markers such as *nfatc1* or *bmp4* in future analyses could further enrich the mechanistic understanding of the phenotype in the AV valve development.

3. Thank you for the thoughtful and well reasoned response. I appreciate the authors' acknowledgment of the importance of identifying early affected genes in elucidating the regulatory mechanisms underlying the heart enlargement phenotype. The technical challenge posed by the small size of the larval heart is understandable. However, recent advances in single-cell and low-input RNA sequencing approaches have made early developmental stage profiling more accessible. For example, the study by Abu Nahia et al. (2024) employed single-cell RNA-seq to successfully dissect cardiac lineage diversity and regulatory pathways involved in heart rhythm during early zebrafish development (*iScience*. 2024 May 22;27(6):110083. doi: 10.1016/j.isci.2024.110083). This work highlights the feasibility of obtaining meaningful transcriptomic data from early-stage hearts, even at single-cell resolution. While I recognize that this may fall outside the current study's scope, the inclusion of such analyses in future work or even exploratory pilot data could significantly strengthen the mechanistic understanding of the phenotype described. I look forward to seeing how the authors may build on this direction in subsequent studies.

Open Access This Peer Review File is licensed under a Creative Commons Attribution 4.0 International License, which permits use, sharing, adaptation, distribution and reproduction in any medium or format, as long as you give appropriate credit to the original author(s) and the source, provide a link to the Creative Commons license, and indicate if changes were

made.

Response to reviewer comments

We sincerely thank the reviewers for their thorough assessment of our manuscript and their valuable comments and suggestions, which have been greatly beneficial to us. Following the reviewers' comments, we have revised the manuscript and added additional data. In the responses below, the reviewers' comments are shown in black **bold**, and our responses are in blue. The added or modified texts are highlighted in yellow in the manuscript. We genuinely hope that the revised manuscript will now be considered suitable for publication in Communications Biology.

Reviewer #1 (Remarks to the Author):

Major issues:

- 1. There is little characterisation of the TG (tomato) used, particularly its genetic background, making the usefulness of these animals as models of heart failure unclear. One of the critical points is to determine the locus where the transgene is inserted and to assess whether this insertion affects levels of gene expression at that locus. The authors showed that removal of DsRed and BGH-pA by Cre recombination rescued the phenotypes. However, the other transgenes, including Tol2 element, tph2 promoter, ChR2, YFP, SV40-pA, remain, so it is unclear why only DsRed and BGH-pA lead to cardiac phenotypes and other transgenes do not.**

We thank the reviewer for the critical suggestions. We have revised the manuscript to address the concerns, including providing essential genetic background information and additional whole genome sequencing data.

First, to provide essential genetic background information, we have added the following sentence in the Methods section: “Tol2-based transgenesis was performed to generate a new transgenic zebrafish line, *Tg(-4.9tph2:LOXP-DsRed express-LOXP-ChR2-YFP); casper (mitfa^{w2/w2}; mpv1^{7a9/a9})* as described previously.⁴⁶ Approximately 25 ng of the plasmid carrying the artificial gene, along with 25 ng of *Tol2* mRNA, were injected into 1-cell stage of *casper* embryos. The injected embryos were raised and maintained in a

heterozygous background by crossing with *casper* (*mitfa*^{w2/w2}; *mpv1*^{a9/a9}). Obtained fish were identified by observing the RFP fluorescence in the zebrafish brain at 3 dpf using an Olympus SZX16 stereomicroscope (OLYMPUS). Zebrafish displaying RFP fluorescence were categorized into the *tomato* group, whereas those lacking RFP fluorescence were classified into the WT group. Supplementary File 1 shows the complete sequence of the plasmid used for the generation of the transgenic line and the structure of the transgene is shown in Supplementary Figure S1A.” (Lines 328–338 in the revised manuscript).

We fully agree that identifying the precise genomic insertion site is a critical issue. We performed whole-genome sequencing (WGS). Through this approach, we have successfully identified an approximately 200 bp genomic sequence adjacent to the breakpoint. However, due to the multimapping nature of this fragment, the precise locus remains unresolved. Despite this limitation, the Mendelian inheritance pattern of DsRed fluorescence supports the presence of a single insertion site. This analysis has been incorporated into the Results section. Given the limitations of short-read sequencing, we acknowledge that long-read sequencing would be better suited for resolving the precise genomic location of the transgene, and we plan to pursue this approach in future studies as resources become available.

We have added the following sentence in the Results section: “To investigate the genomic integration sites of the exogenous transgene, WGS was performed on the *tomato* DNA sample. Sequence reads were mapped to both the zebrafish reference genome (GRCz11) and the full-length plasmid sequence (Supplementary file 1), with high-homology regions masked to minimize false positives. The relative read depth between host and vector sequences at these regions ranged from 6X to 10X, suggesting the presence of multiple transgene copies. Initial SV analysis using automated software pipelines Dynamic Read Analysis for GENomics (DRAGEN; Illumina) did not identify any statistically significant breakpoints indicating transgene insertion because the discordant and split reads were mapped to multiple loci. Supplementary manual inspection with the Integrative Genomic Viewer (IGV)²⁰ identified four discordant and twelve split reads, which all shared a highly identical sequence segment (Figure S7A). Based on this sequence, combined with

PCR, Sanger sequencing, and BLASTN analysis, we identified an approximately 200 bp genomic fragment that is presumably adjacent to the transgene (Supplementary File 2). However, owing to its multimapping nature, the precise genomic insertion site of the transgene remains unresolved.” (Lines 190–203 in the revised manuscript).

Figure S7. Alignment of all whole genome sequencing (WGS) reads. The shared genomic sequence segment among reads is highlighted in blue. Red arrows indicate the breakpoint.

We have added the following sentence in the Methods section: “Genomic DNA was extracted from adult *tomato* zebrafish liver tissue using DNeasy Blood & Tissue Kit (#69504, Qiagen) following the manufacturer’s protocol. Briefly, tissue was homogenized in ATL buffer using Biomasher II (Nippi), followed by proteinase K treatment at 56 °C for 1 h until complete lysis. RNA was removed by RNase A (#19101, Qiagen) treatment. The lysate was then applied to a spin-column, washed with AW1 and AW2 buffers, and eluted in AE buffer. DNA concentration and purity (ratio of the absorbance at 260 and 280 nm ($A_{260/280}$) and $A_{260/230}$) were assessed using a NanoDrop One (ThermoFisher). The DNA sample selected for WGS had a concentration of >80 ng/μl in 50 μl, and met quality thresholds of $A_{260/280} \geq 1.6$ and $A_{260/230} \geq 1.6$.” (Lines 414–422 in the revised manuscript).

“Short-read WGS was performed by Takara Bio Inc. Genomic DNA (2.2 μg) was fragmented using Covaris to obtain DNA fragment of several hundred base pairs.

Genomic DNA libraries were prepared using the TruSeq DNA RCP-Free Library Prep Kit (Illumina) and indexed with IDT for Illumina – TruSeq DNA UD Indexes v2 (Illumina), following the manufacturer’s protocol. The quality and fragment size distribution of the final libraries were evaluated using TapeStation HSD5000 assay (Agilent Technologies).

Sequencing reads were mapped to both the zebrafish reference genome (GRCz11) and the full-length transgene vector sequence using DRAGEN Bio-IT Platform (v4.3.6). To minimize false-positive structural variants (SV), highly similar sequence regions between the genome and transgene were masked before analysis using BLAST (v2.13.0+) and bedtools (maskFastaFromBed, v2.28.0). For SV detection, discordant read pairs and split reads were extracted from alignment files using SAMtools (v1.10) and LUMPY (v0.2.13). Manual inspection of candidate insertion signals was conducted using the Integrative Genomics Viewer (IGV). SV annotation was performed with SnpEff (v4.2) to assess potential functional impacts of the identified breakpoint.” (Lines 425–439 in the revised manuscript).

The insertion itself does not cause a visible phenotype because Cre-mediated removal of DsRed successfully rescued the cardiac defects and the *YFP* gene derived from the vector we used to create the transgene was still detected. The results indicated that the rescued fish still have insertion(s) on their genomic DNA, but the rescued fish showed no obvious defects in the heart. We assume that the differences between *tomato* and Cre-injected fish can be explained by the lower toxicity of YFP protein and/or fewer copies of the *YFP* gene in the genome.

We added the following sentence in the Discussion section: “Despite performing WGS, we were unable to identify the definitive insertion site for the transgene. This is likely due to limitations inherent to short-read sequencing. However, crossbreeding revealed that the inheritance pattern of DsRed followed Mendelian laws, supporting our hypothesis that a single insertion locus exists. This interpretation is further strengthened by Cre recombination experiments, where removal of the DsRed segment while retaining the remaining Chr2-YFP sequence still rescued heart defects. These findings suggest that the cardiac enlargement phenotype is not attributed to insertional disruption of endogenous

genes, but is rather associated with DsRed expression itself. Further investigations using long-read sequencing and targeted functional studies will be required to determine the precise insertion site and clarify the role of DsRed in cardiac valve pathology in *tomato*.” (Lines 260–269 in the revised manuscript).

- 2. The authors mentioned phenotypes in heterozygotes without data that need to be shown in figures. The phenotypic penetrance in the homozygous TG should be determined. They argue that RFP expression in the TG is observed in serotonergic neurons. However, there are no experiments to determine which cell types express RFP by immunostaining or using a reporter fish.**

We thank the reviewer for this valuable comment. To clarify, our study focused exclusively on heterozygous *tomato* fish, as homozygous specimens were not included in the experiments. As mentioned in the first question, the injected embryos were raised and maintained in a heterozygous background.

Moreover, the phenotypic penetrance of dominant heterozygotes is 100%, as all heterozygous *tomato* fish exhibited the expanded heart phenotype, while none of the wild-type fish displayed this phenotype. To address the reviewer’s comment, we have now included the data on heterozygous phenotypes in new Figure S1C, where the frequency of heart enlargement in heterozygous and wild-type fish is quantitatively compared. It is reasonable to conclude that dominant homozygotes would also exhibit 100% phenotypic penetrance.

For more clarity, we have deleted the words “genotype” and “heterozygous” from the manuscript unless strictly necessary. We removed the clause “Zebrafish expressing RFP, regardless of whether the genotype was heterozygous or homozygous,” to avoid confusion. We have also properly replaced "RFP" with “DsRed” in the revised manuscript as required in minor issues.

We have also added the following sentence in the Results section: “In a breeding experiment, *tomato* fish were crossed with wild-type (WT) fish, resulting in offspring with a 1:1 phenotypic ratio, in accordance with Mendelian inheritance. Moreover, all *tomato* fish exhibited heart enlargement, while none of the WT fish showed this

phenotype, confirming that the transgenic insertion is tightly associated with the expanded heart phenotype (Figure S1C).” (Lines 61–65 in the revised manuscript).

Regarding the expression of DsRed in serotonergic neurons, we observed red fluorescence in the brain in 3 dpf zebrafish larvae under a fluorescence microscope. The red fluorescence localization pattern closely matches the *tph2* expression pattern determined by whole mount in situ hybridization (ISH) in 3 dpf WT zebrafish. Since *tph2* expression is known to be localized in the serotonergic neurons of the dorsal raphe nucleus, this localization pattern suggests that the DsRed is expressed in the serotonergic neurons. However, verifying the precise cell types through immunostaining or reporter fish is beyond the scope of this study, as our primary focus is on the impact of DsRed aggregation in the heart.

We have added the following sentence in the Results section: “As expected, transgenic fish carrying this artificial gene exhibited DsRed expression in the dorsal raphe nucleus, in a pattern consistent with *tph2* expression detected by whole-mount in situ hybridization (ISH) (Figure S1B). The transgenic fish line was termed *tomato*.” (Lines 55–58 in the revised manuscript).

Figure S1. B, The DsRed expression pattern resembles the *tph2* expression pattern in the brain at 3 dpf. Schematic representation of DsRed expression under a fluorescence microscope from lateral and dorsal views (left). Diagram of *tph2* expression pattern in the brain detected by whole-mount in situ hybridization (right). White arrows point to DsRed expression, black arrows point to *tph2* expression. Scale bar, 200 μ m. **C,** Schematic diagram of breeding experiment. Black arrows point to zebrafish hearts. Scale bar, 5mm.

Also, we have added the following in the Methods section: “For whole-mount in situ hybridization, 3 dpf WT zebrafish were fixed with 4% paraformaldehyde (PFA). For section in situ hybridization, 3 mpf fish heads and thoraxes were fixed with 4% PFA, followed by demineralization with EDTA-based solution G-chelate mild (GCM-1, Genostaff) for 10 days. The fish were embedded in paraffin and sectioned. In situ hybridization was conducted essentially as previously described. Briefly, *tph2* cDNA corresponding to 498-1051 nt was amplified and cloned into pBluescript II KS. Digoxigenin-labeled RNA probes were synthesized and hybridized at 56 °C for 16 hours, followed by immunodetection using Anti-Digoxigenin-AP, Fab fragments (11093274910, Roche). For color development, 4-nitro blue tetrazolium chloride (NBT) and 5-bromo-4-chloro-3-indolyl-phosphate (BCIP) were used. Imaging was performed using a stereomicroscope (SZ61, OLYMPUS) for whole-mount in situ hybridization and an upright microscope (BX43, OLYMPUS) for section in situ hybridization.” (Lines 401–411 in the revised manuscript).

- 3. They found defects in the atrium and AV valve in the TG hearts and no significant changes in the size of the ventricle. It would be worth to analyse EF and FS in the ventricles of the TG. Given that patients with HF have defects in the ventricles, such as reduced cardiac function, and morphological defects, it is thus not clear whether their TG fish model can be useful to study human HF due to minimal data provided.**

We fully agree with the reviewer that analyzing ejection fraction (EF) and fractional shortening (FS) in the ventricles would provide valuable insights on cardiac function. However, due to technical limitations of the ultrasound imaging system available during

the study, such as probe size, we were unable to obtain sufficiently clear cardiac images. Therefore, we were unable to detect the EF and FS using our current setup (see below).

We acknowledge that our transgenic zebrafish model does not exhibit significant ventricular or myocardial structural changes. However, we observed other signs that directly or indirectly support the presence of heart failure in the transgenic fish, such as an elevated level of the heart failure marker *nppb*, higher mortality, and altered heart rhythms. In addition, we observed clear thickening of the atrioventricular valves, which is a key pathological feature of valvular heart disease (VHD). Our model provides a valuable tool to explore the early pathological changes that drive the progression of valvular pathology into overt heart failure. Moreover, we detected DsRed aggregates in the heart valves. This presents a unique opportunity to investigate the effects of protein deposition and the long-term progression of disease in valvular tissues.

To clarify our interpretation of the pathophysiology of the mutant, we revised the first paragraph of the Discussion as follows: “In the present study, we demonstrated that the novel transgenic zebrafish *tomato* displayed chronic valvular heart disease and heart failure. The transgenic line manifested several characteristics of valvular diseases, presenting both morphological and functional abnormalities of the AV valve. In addition, the elevated mortality rate, reduced heart rate and higher *nppb* expression level in *tomato* suggested the presence of heart failure. In *tomato* fish, evidence of DsRed presence in the valve was detected as early as 14 dpf, which preceded any other cardiac abnormalities.

Cre-loxP-mediated removal of DsRed successfully rescued the phenotype, implicating DsRed as a key driver in the development of the cardiac defect. Collectively, we reported a DsRed-induced, progressive model of cardiac dysfunction that manifests characteristics of valvular dysfunction and heart failure.” (Lines 206–215 in the revised manuscript).

To explain how our transgenic fish model can be useful to study human heart failure, we have added the following sentences in the Discussion part: “Zebrafish have also been utilized in research on cardiac valves¹⁰; however, most research has heavily focused on the developmental processes^{8,36} and phenotypes present in embryonic to early larval stages.^{37,38} Our current model exhibited no overt apparent cardiac abnormality during the embryonic stages (approximately 72 hpf); however, a progressive chronic cardiac phenotype emerged following the appearance of DsRed in the AV valve at 14 dpf. At 30 dpf, cardiomegaly became prominent, reflecting the atrial enlargement. Heart enlargement progressed by 3 mpf, at which point AV valve thickening, regurgitation, and an elevated level of *nppb* expression were confirmed. This series of cardiac phenotypes suggests a possible pathology of the model; first, DsRed accumulates in the AV valve, leading to the valve’s thickening and insufficiency. This valvular dysfunction then causes atrial extension by excessive blood volume and possibly heart failure. The progressive cardiac phenotype of *tomato* is distinct from previously reported cardiac disease models in zebrafish that show signs of heart failure^{14,39,40} or AV valve dysfunction^{37,38} during the early stages of life. In addition, the increase in mortality rate in *tomato* is lower than in the existing models. Many adult *tomato* fish survive for >1 year, which is comparable to the WT. This extended lifespan makes them ideal models for investigating aging heart failure. Moreover, *tomato* can easily be distinguished from their WT littermates by simply observing DsRed fluorescence in the dorsal raphe nucleus. This non-invasive identification method allows transgenic line detection as early as 3 dpf. Therefore, it enables researchers to observe the progression of heart failure and valvular lesions from early larval stages. Taken together, *tomato* could serve as a powerful cardiac disease model that mimics an aspect of human heart diseases that was not illustrated by the existing model.” (Lines 275–295 in the revised manuscript).

Based on this point, we have added the following paragraph in the Discussion section to discuss both the limitations and strengths: “The mechanism through which valvular defects contribute to heart failure in zebrafish remains unclear. However, given that valvular diseases such as tricuspid or mitral regurgitation are well known to contribute to heart failure in human^{41,42}, investigating this relationship in zebrafish remains biologically relevant. We acknowledge that our study provides limited functional evidence; however, the cardiac phenotypes observed in the *tomato* line suggest that it may reflect certain aspects of the valvular disease-induced heart failure. In addition, the methodology to quantify cardiac function in juvenile and adult zebrafish has not been established, except for performing echocardiography requiring specific equipment.⁴³ Although these knowledge gaps and technical restraints remain a possible limitation in the present study, we expect that our transgenic line could serve as a model to further study zebrafish valvular dysfunction and heart failure in their adulthood.” (Lines 298–308 in the revised manuscript).

- 4. The authors show cardiac phenotypes only in adult TG, but it is also important to investigate phenotypes in embryos and determine when heart size, cardiac function, AV morphology, and nppb levels are affected.**

We are grateful for the reviewer’s constructive feedback. Long-term continuous observation from the larval stage is indeed valuable. In response to this suggestion, we have conducted additional analyses of heart size and cardiac function in larvae at different stages. The results revealed that the atrial size of *tomato* was observed to be larger than that of WT in 1 mpf and 2 mpf (Figure S2A and S2B). However, there was no significant difference in heart rate between *tomato* and WT in 3 dpf and 2 mpf (new Figure S3A). We have revised the following sentence in the Results section: “Among these, heart rate serves as an indicator of the overall cardiac function. We analyzed heart rate changes at different developmental stages and found no significant difference in heart rate at the larval period (3dpf) and juvenile stage (2mpf) (Figure S2A and S2B). Although the result indicated a significant decrease in adult (3mpf) *tomato* fish, averaging 91 beats per minute, compared with 104 beats per minute in WT fish (Figure

2B). This was consistent with previous findings indicating lower heart rate in heart failure models of zebrafish.¹⁴⁻¹⁶ (Lines 75–81 in the revised manuscript).

Figure S2. **A**, Heart rates measured in WT and *tomato* fish at 3 days post-fertilization (dpf). N=20 fish in each group. BPM, beats per minute. **B**, Heart rates measured in WT and *tomato* fish at 2 months post-fertilization (mpf). N=15 fish in each group. Data are presented as mean \pm SEM. Statistics: unpaired 2-tailed student t test in (A,B).

“To determine the developmental origins of these cardiac abnormalities, the hearts of both WT and *tomato* zebrafish were extracted and compared at multiple developmental stages (1 mpf, 2 mpf and 3 mpf). The most notable difference was atrial enlargement, evident as early as 1 mpf and sustained throughout subsequent developmental stages (Figure 3A and Figure S3).” (Lines 97–101 in the revised manuscript).

Figure S3. Representative images of hearts extracted from wild-type (WT) and *tomato* zebrafish at 1 month post-fertilization (mpf) and 2 mpf. White dotted lines: atrium. Green lines: ventricle. Scale bars, 500 μ m. A: atrium, V: ventricle.

However, due to the small size of the larval heart and technical challenges in extracting sufficient mRNA from it, we were unable to assess AV morphological structure or quantify *nppb* expression changes in the larvae.

- 5. They performed RNA-seq analysis in 3 mpf fish, which have already developed phenotypes, making it difficult to study the direct effects of the plasmid insertion leading to AV phenotypes.**

We appreciate the reviewer's insight on this point. Analyzing mRNA sequencing results from the larval heart would indeed help in studying the impact of the plasmid insertion on the AV valve phenotype. However, due to the extremely small size of the larval heart, particularly in WT fish, it is technically challenging to isolate sufficient RNA for sequencing. Given these limitations, we were unable to perform RNA-seq in larvae, but we recognize its importance and aim to address this in future work as methods improve.

Minor issues:

- 6. The name of the TG needs to be in accordance with the rules of ZFIN (Zfin.org) so that people in the field can obtain much information about the type of TG.**

According to the reviewer's comment, we have revised the *tomato* line as *Tg(-4.9tph2:LOXP-DsRed express-LOXP-ChR2-YFP); Casper(mitfa^{w2/w2}; mpv17^{a9/a9})*. For detailed modifications in the manuscript, please refer to our response to Reviewer #1Q1.

We have also altered the presentation of the casper fish as "*casper (mitfa^{w2/w2}; mpv17^{a9/a9})*."

In addition, we have changed "MCM4, PCNA, and TRIP13" to "*mcm4, pcna, and trip13*" in accordance with the ZFIN's guidelines.

- 7. Using 'Tomato', 'RFP', and 'DsRed Express' interchangeably may lead to confusion.**

We appreciate reviewer's feedback. In response to your comment, we have properly replaced "RFP" with "DsRed" in the revised manuscript.

8. In Figure S1 A, non-English language is used.

Thank you for the reviewer's astute observation and valuable feedback regarding the presentation. We have modified and replaced Figure S1A.

Reviewer #2 (Remarks to the Author):

Major concerns.

1. It is unclear how many insertions of the construct are in the "tomato" mutant

We thank the reviewer for raising this important point. Based on the whole-genome sequencing result, we estimate that there are approximately 6 to 10 copies of the construct present in the *tomato* mutant. A detailed response addressing this point can be found in our reply to Reviewer #1Q1.

2. The authors make a rescue experiment with injecting Cre mRNA in heterozygous fish. They briefly describe that tomato heterozygous embryos as showing "an expanded heart" but do not present or analyze further the heterozygous phenotype and how similar it is to the homozygous phenotype. It seems that the genetics of the heart failure phenotype are recessive (page 11, figure 2A), which makes it unlikely to be an RFP protein aggregation phenotype (that would be dominant) and points to a locus specific disruption of a gene. The genomic context of the insertion(s) should be identified by flanking DNA sequencing and presented.

We thank the reviewer for this valuable comment and the opportunity to clarify a possible misunderstanding. Our study focused exclusively on heterozygous *tomato* fish, as homozygous specimens were not included in the experiments. The injected embryos were raised and maintained in a heterozygous background. Please see our response to Reviewer #1Q2, who had similar comments.

In regards to the genetic nature of the phenotype, we respectfully disagree with the interpretation that the heart defect is recessive. As mentioned in Figure S1B, heterozygous transgenic fish carrying this artificial gene exhibited DsRed expression, whereas wild-type fish did not, supporting this point. Additionally, our preliminary data following Cre-injection demonstrated that the DsRed expressing gene was successfully excised from the genome, while the YPF gene, originating from the vector used to

construct the transgene, remained detectable. This supports the interpretation that the observed phenotype is not due to a locus-specific disruption of an endogenous gene.

Regarding the locus where the transgene is inserted, as described in our response to Reviewer #1Q1, we have made efforts to identify transgene insertion through whole-genome sequencing. Specifically, we identified an approximately 200 bp genomic sequence adjacent to the breakpoint. However, due to the highly multimapping nature of this fragment, we could not confirm the exact insertion site. Nonetheless, our findings provide partial information about the genomic sequence, and we continue to work toward resolving the exact genomic locus.

- 3. It is unclear if the DsRed protein they see in the adult valves is driven by the physiological expression of the *thp2* gene. An in situ should at these stage should be performed to show if this kind of neurons exist in the valves or elsewhere in the adult heart.**

We sincerely appreciate the reviewer's suggestion. We have performed section in situ hybridization to investigate *tph2* expression in the heart. Our results showed that the *tph2* expression in the atrioventricular valve was low and was at a level similar to the background tissues.

We have added the following sentence in the Results section: "To examine whether the DsRed expression is driven by the physiological expression of *tph2* in the valve, section in situ hybridization (ISH) was performed (Figure 5D). Although the *tph2* was expressed in the AV valve, the expression level was low and not noticeably different from that observed in surrounding cardiac and other tissues." (Lines 174–177 in the revised manuscript).

Figure 5. D, *tph2* expression in AV valve region of two WT zebrafish was detected by section in situ hybridization at 3 mpf. Black dotted lines: atrioventricular (AV) valves. Scale bar, 50 μ m.

We have also modified the following sentence in the Discussion section: “Since the transgenic component was designed to express DsRed under the control of the *tph2* promoter, we first predicted that DsRed expression would be driven by the physiological expression of *tph2* in the valve. Contrary to expectations, in situ hybridization revealed a low-level expression of *tph2* in the AV valve, similar to background levels. This result suggests that the presence of DsRed cannot be solely attributed to the physiological expression of *tph2* in the valve, implying other valve-specific mechanisms that contribute to the phenotype.” (Lines 254–259 in the revised manuscript).

We have also deleted the following part in the Abstract: “These findings advance our understanding of the effects of TPH2 on heart valves.”

4. Overall the immunohistochemistry pictures are of low resolution and the heart failure phenotype not very well documented and supported. The atrium looks indeed dilated but what about the cardiomyocyte phenotype? Is there an atrial specific hyperplastic response or are the atrial cardiomyocytes bigger? And what about the ventricular phenotype (ventricles appears to be smaller in the mutants, is there apoptosis or any other explanation for this phenotype?) . Higher magnification

figures of adult cardiomyocyte size/shape in atrium and ventricle should be included.

We sincerely appreciate the reviewer's suggestion to comment on our data. We have replaced the original immunohistochemistry images with higher-resolution figures to improve clarity and better illustrate the observed structural changes.

Regarding the atrial phenotype, our data did not reveal evidence of cardiomyocyte hypertrophy. Instead, the enlarged atrium appears to result from passive dilation, caused by excessive blood retention rather than an intrinsic hyperplastic or hypertrophic response.

We conducted quantitative analysis, and according to Figure 3C, there is no significant difference in ventricular area between wild-type and *tomato* zebrafish.

We have added the following sentence in the Results section: "In addition, hematoxylin and eosin (H&E) staining revealed disorganized and loosely arranged myocardial fibers in the *tomato* atria, indicating structural remodeling and potential functional impairment (Figure S4). No obvious morphological alterations were observed in the ventricular myocardium." (Lines 113–116 in the revised manuscript).

Figure S4. Magnified H&E stained images of atrial and ventricular myocardium in adult zebrafish. Representative coronal heart sections from WT (A-C) and *tomato* (D-F). **A**, Overview of the WT heart. Boxed areas corresponding to higher magnification images of ventricle (**B**) and atrium (**C**). **D**, Overview of the *tomato* heart. Boxed areas corresponding to higher magnification image of ventricle (**E**) and atrium (**F**). Scale bar, 200μm in **A** and **D**; 20μm in **B**, **C**, **E** and **F**. A: atrium, V: ventricle.

- 5. The valves presented in Figure 3H-I do not appear to be at comparable positions, which admittedly the atrial dilation makes it very tough. Could they make a compilation of the other 3 fish they measured (wt and tomato) as a supplementary figure, so one can appreciate the differences in thickness.**

We greatly appreciate your constructive comment. To address this, we have added a supplementary figure showing atrioventricular (AV) valve morphology from additional WT and *tomato* zebrafish, allowing for more comprehensive comparison of valve thickness across specimens. We would like to note, however, that due to the method of

serial sectioning, the thickest region of the valve may not be included in the sections selected for H&E staining across all samples. In some cases, the optimal section containing the thickest part of the valve may have been used for other analyses, such as immunostaining, and was therefore not available for morphological comparison. As a result, while not all images may represent the absolute maximum thickness, the overall trend, a significant increase in valve thickness in *tomato* fish compared to WT, remains clear.

Figure S5. Magnified images of AV valve from additional 3 samples in WT and *tomato* zebrafish, corresponding to sample shown in Figure 3H-I. Dark blue dotted lines: atrioventricular (AV) valves. Scale bar, 50 μ m. A: atrium, V: ventricle.

Reviewer #3 (Remarks to the Author):

Comments:

- 1. In the result, the authors states that no distinct phenotypes were observed in the mutants during the embryonic phase. However, as tomato fish aged, they unexpectedly developed expanded hearts. On contrary, in the introduction, they have mentioned that severe heart enlargement was observed during the larval stage, which increased with age. Please clarify.**

We sincerely appreciate your attention to detail. We refer to fish from the one-cell stage to 72 hpf as “embryos” and fish from 3 dpf to 6 weeks post fertilization as “larvae.” We have now added the actual age in the revised manuscript for clarity. We have revised this sentence “Heart enlargement was observed during the larval stage (30 dpf) and became more pronounced with age.” (Lines 37–38 in the revised manuscript).

- 2. Did the authors perform heart rate analysis in the developmental stages? They should present this analysis to support their claim.**

We appreciate the reviewer’s insightful suggestion regarding the data of heart rate analysis in the developmental stages. We have conducted additional analyses heart rate at different stages. There was no significant difference in heart rate between tomato and WT at 3 dpf and 2 mpf (new Figure S3A). Please see our response to Reviewer #1Q4, who had similar comments.

- 3. The validation of gene expression for at least the highlighted genes in each pathway for example the inflammation etc. will be necessary to confirm the finding.**

We understand the reviewer’s request. We performed qPCR analysis to validate gene expression across multiple pathways. In the inflammation pathway, we analyzed *marco*, *cybb* and *il7r*. and found that *marco* and *cybb* were significantly upregulated in *tomato* zebrafish. For the EMT pathway, we examined *tspi2*, *bmp6* and *fossa*, all of

which showed significantly elevated expression levels in *tomato* zebrafish. Similarly, in the TNF- α -NF κ B pathway, *cd44a* and *nfk2* exhibited increased expression levels in tomato zebrafish. Finally, in the IL-2 STAT5 signaling pathway, *sox8a* showed a significant upregulation, whereas *tbx1* exhibited no notable change. These findings further support our RNA seq result and highlight the key pathways affected in tomato zebrafish.

Figure S6. **A**, Relative expression of *marco*, *cybb*, *il7r* in inflammatory response. **B**, Relative expression of *tfpi2*, *bmp6*, *fossa* in Epithelial mesenchymal transition. **C**, Relative expression of *tbx1* and *sox8a* in IL-2-STAT5 pathway. **D**, Relative expression of *cd44a*, *nfk2* in TNF α signaling via NF κ B pathway. The expression level of mRNA from WT and *tomato* fish hearts at 3mpf were normalized to *actb1*. N = 5 fish in each group. Data are presented as mean \pm SEM. Statistics: Multiple unpaired t test with the Holm-Sidak correction in (A–D).

We have added the following sentence in the Results section: “To validate the RNA sequencing results, we performed qPCR analysis targeting representative genes from key pathways. Inflammatory genes *marco* and *cybb* (Figure S6A), EMT-related marker *tfpi2*, *bmp6* and *fossa* (Figure S6B), *sox8a* in IL-2-STAT5 pathway (Figure S6C), as well as TNF α -NF κ B targets *nfk2* (Figure S6D), all showed significant upregulation in *tomato* zebrafish.” (Lines 141–145 in the revised manuscript).

We have also revised the Methods section accordingly:

“RNA extraction and quality assessment

The extracted hearts were dissected and kept in RNAlater (#AM7021, ThermoFisher) solution. Five samples were prepared for WT and *tomato*, and two hearts were pooled per sample. Total RNA was extracted from the cell suspension using RNeasy Fibrous Tissue Mini Kit (#74704, Qiagen) following homogenization in buffer RLT. RNA quality was assessed using RNA ScreenTape (#5067-5576, Agilent Technologies), with the TapeStation 4150 system (Agilent Technologies). RNA samples with an RNA Integrity Number Equivalent (RINe) >8 were used for further analyses. The RNA samples were stored at -80°C until use.” (Lines 441–448 in the revised manuscript).

“Quantitative PCR

Total RNA was reverse transcribed into complementary DNA using the High-Capacity RNA-to-cDNA kit (#4387406, ThermoFisher). qPCR was performed using Applied Biosystems QuantStudio 6 Flex and QuantStudio 3 (Applied Biosystems) with SsoAdvance Universal SYBR Green Supermix (#1725271, BioRad). In this research, the target genes from PrimePCR™ SYBR Green Assays (Bio-Rad) were: matrix metalloproteinase-2 (*mmp2*) (qDreCID0004788), *mmp9* (qDreCID0002041), *mmp13a* (qDreCED0018863), collagen, type I, alpha 1a (*colla1a*) (qDreCED0017616), *colla2* (qDreCED0019386), *nppb* (forward 5' TGTTTCGGGAGCAAACCTGGA 3' and reverse 5' GTTCTTCTTGGGACCTGAGCG 3'), *marco* (qDreCED0010322), *cybb* (qDreCED0014759), *il7r* (qDreCID0005046), *tfpi2* (qDreCID0002865), *bmp6* (qDreCID0014051), *fosas* (qDreCED0015103), *cd44a* (qDreCID0001835), *nfkb2* (qDreCID0020554), *tbx1* (qDreCED0015105), *sox8a* (forward 5' AAACCTCGCCGATCAGTACCC 3' and reverse 5' TGCAGCCTCAGTCTTTCAGC 3'), *tph2* (qDreCED0017433) and the housekeeping gene *actb1* (qDreCED0020462). The relative gene expression levels were analyzed using the $2^{-\Delta\Delta Ct}$ method. The delta ($\Delta\Delta Ct$) values were normalized to the $\Delta\Delta Ct$ of *actb1*. Each sample was analyzed in either triplicate or duplicate.” (Lines 450–465 in the revised manuscript).

- 4. In the supplemental video files the heartbeat in mutant appeared faster than the wt. please check.**

Thank you for your careful observation and valuable feedback regarding the videos. Upon review, we recognized the need to replace the original WT video to ensure accurate representation. We have now updated the supplemental materials with new recordings under improved conditions (New video S1_wt brightfield and video S2_wt RFP). We appreciate the opportunity to correct this issue.

- 5. A brief information about the methodology used to construction of the transgenic will be helpful in reproducibility.**

To address the reviewer's comment, we have added brief information about Tol2-based transgenesis in the Methods section (Lines 328–338 in the revised manuscript). For detailed modifications in the manuscript, please refer to our response to Reviewer #1Q1.

- 6. The authors mentioned that they measured the weight and length of each fish analyzed. further they also mention about the genders. However, there are no results presented related to this analysis. What is the significance of these variables?**

Thank you for bringing this to our attention. The mention of weight, length, and sex was part of our routine experimental documentation but was not directly relevant to the analyses presented in this study. To avoid confusion, we have removed the corresponding sentence from the Methods section in the revised manuscript.

- 7. The authors should analyze the expression of marker genes for the AV valve development to support the AV thickness observation.**

We thank the reviewer for this insightful comment. We agree that analyzing the expression of marker genes could provide additional context for the observed thickening of the atrioventricular (AV) valve. Certain molecular programs active during valve development may be engaged in pathological remodeling. For instance, as mentioned in the manuscript, vimentin – a marker of epithelial-to-mesenchymal transition (EMT) – plays a crucial role in early endocardial cushion formation and persists in mesenchymal – like valve interstitial cells during both development and disease. Similarly, PCNA, a marker for proliferating cells, is expressed during valve progenitor cell expansion in development and also reflects heightened proliferative activity during abnormal valve thickening.

However, the mechanisms underlying normal valve development and those contributing to abnormal, pathological thickening also remain different. Therefore, the expression of additional developmental marker genes may not be suitable to explain or confirm the pathological phenotype observed in our model. Instead, we believe that a more targeted analysis focusing on pathways associated with valvular remodeling and inflammatory response will be more informative in understanding the basis of the observed valve thickening. In the original manuscript, we have already discussed the contribution of pathways related to valvular cell proliferation, differentiation, and immune-related stimuli to the observed valve thickening.

8. Identification of the early affected set of genes be helpful to uncover the regulatory role in the onset of the heart enlargement phenotype.

Thank you for your insightful comment. Identifying the early affected genes from the larval heart would indeed help in studying the regulatory role in the heart enlargement phenotype. However, the small size of the larval heart currently makes it particularly challenging to obtain sufficient RNA for transcriptomic analysis. Nevertheless, we recognize the importance of this question and aim to address it in future studies as more sensitive techniques become available.

9. An expression analysis of *tph2* to investigate its expression in the heart valve will be helpful. In-situ hybridization would be a feasible approach.

We sincerely appreciate the reviewer's suggestion. We have performed section in situ hybridization to investigate *tph2* expression in the heart. Our results confirmed the presence of *tph2* signal in atrioventricular valve that was comparable to the background. Please see our response to Reviewer #2Q3, who had similar comments.

10. Does the sequencing analysis reveal expression of *tph2* in the mutant? It will be interesting to see how the expression of *tph2* or related genes changes in mutant.

Thank you for your insightful comment. We examined the *tph2* expression using both RNA sequencing analysis and qPCR. Our analysis confirmed that *tph2* is indeed expressed at a low level in the heart. However, we did not observe any significant differences in its expression between wild-type and mutant zebrafish.

We have added the following sentence in the Results section: “Meanwhile, qPCR confirmed that *tph2* expression levels did not significantly differ between WT and *tomato* zebrafish (Figure S6F).” (Lines 177–178 in the revised manuscript).

Figure S5. F, Relative expression of *tph2*. The expression level of mRNA from WT and *tomato* fish hearts at 3mpf were normalized to *actb1*. N = 5 fish in each group. Data are presented as mean ± SEM. Statistics: unpaired 2-tailed student t test in (F).

11. To assess heart rate, zebrafish were anesthetized in 0.2 mg/ μ L tricaine, while Zebrafish were anesthetized in 0.016% tricaine solution diluted in system water for 1 min for ultrasound. Why the doses of tricaine different in these two methods in the same study?

We thank the reviewer for carefully assessing the manuscript, and we apologize for this oversight. While the way we prepared tricaine was consistent, we applied different measurement units based on those used in the relevant experimental literature. We have standardized both concentrations to 0.016% in the revised manuscript.

Additionally, we have changed some parts of the manuscript to match the style of *Communications Biology*.

First, we have moved the Results section right after the introduction section. Then, the Method section moved to the last part of the manuscript. Accordingly, we adjusted our reference numbers and included the full spelling of the abbreviated terms at their first mention.

We have added the following sections:

“Data availability

The source data for all statistical graphs are provided in Supplementary Data 1. Additional information is available from the corresponding author upon reasonable request.”

“Author contributions

S.M. established the zebrafish model. S.M. and R.K. together conceived the project. Y.T.L. and S.M. conducted the experiments and analyzed the data. S.R.S. and T.N. helped with technical support. Y.T.L. and S.M. wrote the first version of manuscript. R.K. and M.H. supervised the project. S.M., R.K., S.R.S., T.N., and M.H. reviewed and edited the manuscript. All authors read and approved the final manuscript.”

We have deleted the following sections:

“Short title: Chronic heart failure zebrafish model”

“Key words: zebrafish, chronic heart failure, red fluorescent protein, cardiac valve”

“Nonstandard Abbreviations and Acronyms

AV	atrioventricular
BPM	beats per minute
CHF	chronic heart failure
dpf	days post-fertilization
EMT	epithelial-to-mesenchymal transition
FDR	false discovery rate
mpf	months post-fertilization
NPPB	natriuretic peptide B
PCNA	proliferating cell nuclear antigen
PCA	principal component analysis
RFP	red fluorescent protein
TPH2	tryptophan hydroxylase 2
VIC	valve interstitial cell
YFP	yellow fluorescent protein”

“Supplemental Material

Tables S1 and S2

Figures S1–S6

Videos S1–S4

Files S1 and S2”

We have also made some minor alterations to increase clarity and readability, which did not affect the content of our manuscript.

“the red fluorescent protein” (Line 2 in the revised manuscript).

“valvular heart diseases and” (Line 13 in the revised manuscript).

“cardiac structural” (Line 16 in the revised manuscript).

“Among these underlying causes, valvular cardiac disease is a significant contributor, primarily by causing progressive volume and pressure overload that compromises cardiac output performance.²⁻⁴ This maladaptive process is characterized by a

protracted and progressive trajectory, often taking years to transition from the compensated cardiac function to overt heart failure.⁵ Despite significant clinical relevance, the molecular and structural changes driving valvular dysfunction-induced heart failure remain poorly understood.^{5,6} Establishing appropriate animal models that mimic valve-induced cardiac pathology is essential for elucidating the mechanisms of CHF progression and for evaluating potential therapeutic interventions.” (Lines 17-25 in the revised manuscript).

“that expresses the red fluorescent protein DsRed in the cardiac valve, which leads to multiple cardiac phenotypes.” (Lines 36-37 in the revised manuscript).

“valvular heart disease and” (Line 309 in the revised manuscript).

“After assessing the DsRed expression, DsRed-negative wild-type (WT) fish and DsRed-positive *tomato* were placed into separate tanks.” (Lines 347-348 in the revised manuscript).

“Total” (Line 451 in the revised manuscript).

“Libraries for RNA sequencing were prepared by Macrogen Japan Co. as previously described.⁴⁵ A total of 200 ng of RNA was used for library preparation using the TruSeq Stranded mRNA LT Sample Prep Kit Set A (Illumina). The sequencing of the library was performed using the NovaSeq 6000 sequencing system (Illumina) with paired-end 101-bp reads.” (Lines 470–473 in the revised manuscript).

We have added the following sentence to the Acknowledgement section to show our gratitude to the Center for Medical Research and Education at the University of Osaka for their help during our revision process.

“We also thank the Center for Medical Research and Education, Graduate School of Medicine, the University of Osaka for their support.”

Response to reviewer comments

Reviewer #1 (Remarks to the Author):

The authors have addressed all questions, so there are no more concern.

We sincerely thank the reviewer for the positive evaluation of our revised manuscript and for the helpful comments provided during the review process.

Reviewer #3 (Remarks to the Author):

We sincerely thank the reviewer for carefully evaluating our revised manuscript and for providing insightful comments and suggestions. In the responses below, the reviewer's comments are shown in black **bold**, and our responses are in blue. The corresponding changes in the manuscript are highlighted in **yellow**. We are grateful for the reviewer's thoughtful feedback.

- 1. The presence of multiple insertions of a construct (e.g., six copies as in your tomato zebrafish mutant) can significantly impact the interpretation of the mutant phenotype and its linkage to specific genes. For instance, each insertion can disrupt different genomic loci, potentially affecting the expression or function of multiple genes. This complicates the attribution of phenotype to a single insertion or gene. The observed phenotype may be a composite of several disruptions. Please discuss this in your discussion.**

We thank the reviewer for this important comment. Although we could not completely exclude the possibility that multiple different insertion sites exist, the inheritance pattern of DsRed strongly supports the presence of a single insertion locus, which presumably contains multiple (possibly 6-10) copies.

We have revised the Discussion to clarify this interpretation and to acknowledge the reviewer's concern. **"In combination with the relative read depth, the most plausible explanation is that this locus harbors multiple (approximately 6–10) tandem copies of the transgene."**

- 2. I believe incorporating additional markers such as *nfatc1* or *bmp4* in future analyses could further enrich the mechanistic understanding of the phenotype in the AV valve development.**

We sincerely appreciate this constructive suggestion. We agree that incorporating additional markers such as *nfatc1* and *bmp4*, which are well recognized for their roles in valve development and regeneration, could further enhance the mechanistic understanding of AV valve remodeling in our model. While such analyses were not performed in the present study, we will carefully consider them in future work to provide deeper mechanistic insights.

- 3. Thank you for the thoughtful and well reasoned response. I appreciate the authors' acknowledgment of the importance of identifying early affected genes in elucidating the regulatory mechanisms underlying the heart enlargement phenotype. The technical challenge posed by the small size of the larval heart is understandable. However, recent advances in single-cell and low-input RNA sequencing approaches have made early developmental stage profiling more accessible. For example, the study by Abu Nahia et al. (2024) employed single-cell RNA-seq to successfully dissect cardiac lineage diversity and regulatory pathways involved in heart rhythm during early zebrafish development (iScience. 2024 May 22;27(6):110083. doi: 10.1016/j.isci.2024.110083). This work highlights the feasibility of obtaining meaningful transcriptomic data from early-stage hearts, even at single-cell resolution. While I recognize that this may fall outside the current study's scope, the inclusion of such analyses in future work or even exploratory pilot data could significantly strengthen the mechanistic understanding of the phenotype described. I look forward to seeing how the authors may build on this direction in subsequent studies.**

We are grateful for this thoughtful suggestion and for pointing us to the valuable work of Abu Nahia et al. (2024). We fully agree that applying single-cell and low-input RNA sequencing approaches at early developmental stages could provide important insights into the early regulatory mechanisms underlying the heart enlargement phenotype. Although such analyses are beyond the scope of the current study, we highly appreciate this perspective and

will carefully consider incorporating these advanced approaches in future investigations to strengthen the mechanistic understanding of our model.